# Nonvolatile ferroelectric domain wall memory integrated on silicon

Haoying Sun[1,2,5], Jierong Wang[1,2,5], Yushu Wang[1,2], Changqing Guo[3], Jiahui Gu[1,2], Wei Mao[1,2], Jiangfeng Yang[1,2], Yuwei Liu[1,2], Tingting Zhang[1,2], Tianyi Gao[1,2], Hanyu Fu[1,2], Tingjun Zhang[1,2], Yufeng Hao [1,2], Zhengbin Gu[1,2], Peng Wang [3], Houbing Huang [4] & Yuefeng Nie [1,2✉]

Ferroelectric domain wall memories have been proposed as a promising candidate for nonvolatile memories, given their intriguing advantages including low energy consumption and high-density integration. Perovskite oxides possess superior ferroelectric prosperities but perovskite-based domain wall memory integrated on silicon has rarely been reported due to the technical challenges in the sample preparation. Here, we demonstrate a domain wall memory prototype utilizing freestanding $BaTiO_3$ membranes transferred onto silicon. While as-grown $BaTiO_3$ films on (001) $SrTiO_3$ substrate are purely $c$-axis polarized, we find they exhibit distinct in-plane multidomain structures after released from the substrate and integrated onto silicon due to the collective effects from depolarizing field and strain relaxation. Based on the strong in-plane ferroelectricity, conductive domain walls with reading currents up to nanoampere are observed and can be both created and erased artificially, highlighting the great potential of the integration of perovskite oxides with silicon for ferroelectric domain wall memories.

[1] National Laboratory of Solid State Microstructures, Jiangsu Key Laboratory of Artificial Functional Materials, College of Engineering and Applied Sciences, Nanjing University, Nanjing 210093, P. R. China. [2] Collaborative Innovation Center of Advanced Microstructures, Nanjing University, Nanjing 210093, P. R. China. [3] Department of Physics, University of Warwick, Coventry CV4 7AL, UK. [4] School of Materials Science and Engineering & Advanced Research Institute of Multidisciplinary Science, Beijing Institute of Technology, Beijing 100081, China. [5] These authors contributed equally: Haoying Sun, Jierong Wang. ✉email: ynie@nju.edu.cn

Ferroelectric domain walls (DWs), the low-dimensional structures that separate differently polarized regions, have been shown to exhibit appealing properties in recent years[1–7], including metallic conduction[8–13], magnetoresistance[14] and photovoltaic behaviors[15–17]. Conductive DWs, mainly appearing as head-to-head (H-H) or tail-to-tail (T-T) DWs[8,18–21], have been hitherto detected in various types of materials, such as proper ferroelectric perovskites[10,18–20,22], non-perovskites[12,23,24] and improper ferroelectrics[13,25,26]. Due to the nanoscale space distribution of DWs and non-destructive switching of polarization states, ferroelectric DWs hold great potential for applications in high-density and low-power non-volatile memories[18,27,28]. For practical industrial applications in complementary metal-oxide-semiconductor (CMOS)-based technology, however, it is highly desired to integrate these DW memories with silicon (Si). To our knowledge, such attempts have only been demonstrated in LiNbO$_3$[29] and $\alpha$-In$_2$Se$_3$[30] and no success has been achieved in ferroelectric perovskite oxides up to date although they exhibit great ferroelectric properties to facilitate the exploitation of DW memory.

Integrating perovskite oxides with Si has long been a challenging task since the high growth temperature and oxidant environment for the growth of high crystalline quality perovskite oxides can easily oxidize the surface of Si wafer, hindering the subsequent high-quality integration of these two types of materials[31–33]. Adopting special growth recipes, SrTiO$_3$ (STO) can be epitaxially grown on Si with high crystalline quality and show ferroelectricity[34]. Using STO as a buffer layer, many ferroelectric oxides like BaTiO$_3$ (BTO), PbTiO$_3$ have also been integrated with Si[35,36]. Nonetheless, these ferroelectric oxides are mostly under compressive strain and exhibit out-of-plane ferroelectricity, while H-H or T-T conductive DWs normally require in-plane polarization components. Recently, high-quality freestanding ferroelectric perovskite oxides have been achieved using water-soluble Sr$_3$Al$_2$O$_6$ (SAO) sacrificial layer[37,38], providing an effective route to integrate high crystalline quality functional oxides, such as BTO[39] and BiFeO$_3$ (BFO)[40], with Si and other technologically important semiconductors. Moreover, without the clamping effect imposed by the substrate, freestanding oxide membranes have exhibited great tunability in strain engineering and dimensionality[39,41–43]. By a combination of strain relaxation and depolarizing field[44–47], the in-plane ferroelectric domain and H-H or T-T DW structures can be induced and may allow the integration of conductive DW memories on Si.

Here, the emergence of in-plane ferroelectricity is first predicted for freestanding BTO membranes by phase-field simulation (Fig. 1b). Motivated by this promising result, we realized the in-plane multidomain configuration in BTO/Si through as-grown BTO/SAO/STO, and thus the integration of conductive DWs in BTO on the Si substrates. The ferroelectric polarization rotation from out-of-plane $c$ axis to in-plane $a$-($b$-) axis in BTO membrane emerges as a combined effect of strain relaxation and depolarizing field during release from STO substrate. The nA-scale reading current and the systematic performance of the "write-read-erase" process on our prototype signifies a promising step forward in achieving Si-based non-volatile conductive DW memory.

## Results

### Anticipation for in-plane ferroelectricity
The polarization distribution of 25 u.c. BTO thin films before and after releasing from the STO substrate is calculated by phase-field simulations (Fig. 1b). The effects of depolarizing field, epitaxial strain (clamped film), and strain relaxation (freestanding membranes) are considered (see Methods). The BTO thin film clamped on STO substrate demonstrates monodomain with $c$-axis downward polarization, in agreement with the previous reports[48]. However, when the epitaxial strain induced by lattice mismatch vanishes along with the separation from STO substrate, the freestanding BTO membrane exhibits a prominent in-plane multidomain structure. Such in-plane ferroelectricity can be a fertile ground for conductive DWs, encouraging us to experimentally apply a freestanding BTO/Si system as a potential prototype for DW memories.

### DW memory prototype fabrication
As-grown BTO films were synthesized on TiO$_2$-terminated (001) STO substrates after growing 6 u.c. SAO buffer layers by molecular beam epitaxy (MBE, see Methods), with the thicknesses ranging from 6 u.c. to 160 u.c. The reflection high energy electron diffraction (RHEED) oscillations and diffraction patterns are displayed in Supplementary Fig. 1[37,49]. High-quality single crystallinity and smooth step-and-terrace surfaces are both confirmed (Supplementary Fig. 1). As schematically shown in Fig. 1c, we fabricated the ferroelectric DW memory prototype by transferring freestanding BTO membranes onto doped Si. As indicated by high-angle annular dark-field scanning transmission electron microscopy (HADDF-STEM) and energy-dispersive X-ray spectroscopy (EDS) measurements (Supplementary Fig. 2), the BTO membrane conformally attaches to Si substrate. The micrometer-size and well-preserved smooth surfaces of BTO membranes promote the subsequent fabrication.

### 90° polarization rotation during release
To guarantee that conductive DWs in BTO/Si would form as we desired, free-standing BTO membranes should be strongly in-plane polarized, as expected by computation. We made systematic PFM measurements to investigate the domain configurations in BTO membranes, since in-plane ferroelectric domain patterns for fully relaxed BTO membranes have never been experimentally observed to our knowledge.

The $c$-axis oriented ferroelectric polarization of epitaxially compressive-strained BTO grown on SAO/STO was confirmed for our samples in the first place[48]. Bistable switching was managed by applying biases of ±20 V. Figure 2a demonstrates the domain configurations for 30 u.c. as-grown BTO acquired after domain switching, including both out-of-plane (vertical PFM) and in-plane (lateral PFM) results. Though the absence of the bottom electrode and the possible static charges may lead to imperfect amplitude contrast after switching[36,50,51], adequate information has still been delivered. The presence of explicit 180° phase contrast in the vertical PFM (VPFM) phase image together with the absence in the lateral PFM (LPFM) phase image indicates as-grown BTO being purely out-of-plane polarized, which is further affirmed by the out-of-plane phase hysteresis and butterfly-shaped amplitude loop in Fig. 2b. In addition, in VPFM phase image, the unbiased area exhibits evident intrinsic monodomain structure. The consistency between the outermost unbiased area and the innermost positive-biased area further illustrates the existence of a built-in field, which points down to the STO substrate and brings about spontaneous out-of-plane downward polarization in BTO layers, consistent with the previous report[48].

The same domain writing process was then performed on 30 u.c. freestanding BTO membrane transferred onto the Si (the BTO/Si heterostructure), with bias voltages of ±13 V. The VPFM images and LPFM images obtained after switching are respectively shown in Fig. 2c. For VPFM, the created phase contrast is barely visible, implying the weak out-of-plane ferroelectric ground state of relaxed BTO membranes. The faint phase

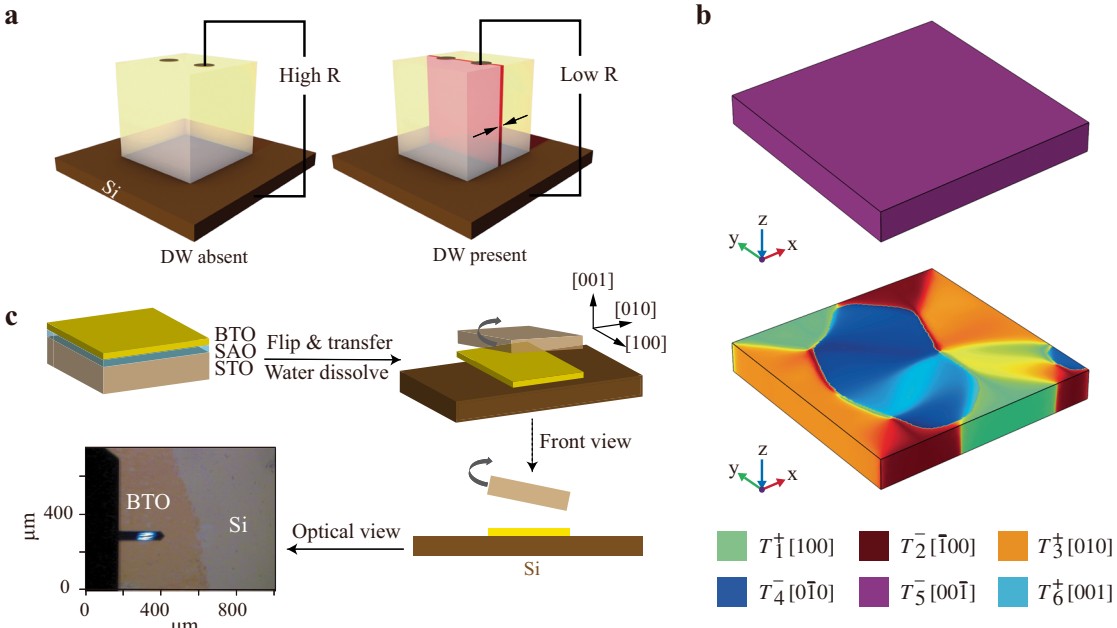

**Fig. 1 Schematics of DW memory prototype and fabrication process. a** OFF (high resistance, left panel) and ON (low resistance, right panel) states for conductive DW memory prototype. Arrows represent polarization orientations and the red interface denotes the position of DW. **b** Phase-field simulation of BTO thin films. Simulated domain structure of as-grown 25 u.c. BTO thin film fully strained on the STO substrate showing pure downward (point to STO substrate) out-of-plane polarization (top panel). Simulated domain structure of a fully relaxed 25 u.c. freestanding BTO membrane showing prominent in-plane domain structures (bottom panel). **c** Schematics of operations for transferring the as-grown BTO film directly onto the Si. The optical image shows the released 30 u.c. BTO membrane on Si substrate.

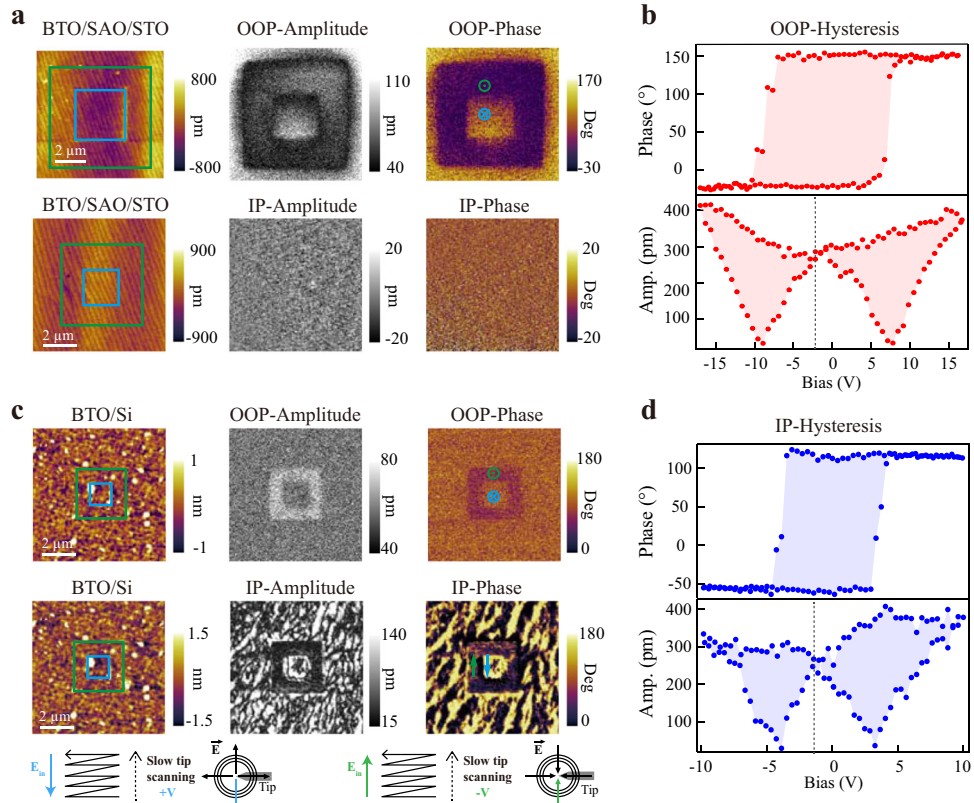

**Fig. 2 Ferroelectric evolution of BTO before and after transferring. a**, **c** Vertical (out-of-plane) and lateral (in-plane) PFM domain switching results for as-grown 30 u.c. BTO/SAO/STO and 30 u.c. BTO/Si. Bias voltages of ±20 V and ±13 V are applied in **a** and **c**, respectively. The blue box and green box areas are scanned by a dc bias tip with positive and negative voltages, respectively. Schematics below **c** illustrate the trailing field method for in-plane polarization switching[54,55]. **b**, **d** Local phase hysteresis loops and amplitude butterfly curves obtained from out-of-plane signal of 30 u.c. BTO/SAO/STO in **a** and in-plane signal of 30 u.c. BTO/Si in **c**, respectively.

contrasts persist in freestanding BTO membranes of thicknesses lower than 30 u.c. (Supplementary Fig. 3a), sometimes even vanishing about an hour after switching (Supplementary Fig. 3b). However, thick BTO membranes, such as 160 u.c. BTO, show sharp phase contrast in poled regimes (Supplementary Fig. 3c), consistent with previous observations[39]. This instability of out-of-plane polarization in thinner freestanding BTO membranes can be understood as a result of the size effect due to strain release and the existence of depolarizing field. The phase matching of the unpoled perimeter and positively poled regime in BTO/Si sample further suggests the intrinsic polarization oriented downwards to Si. Recalling that samples were upended onto Si, the seemingly same downward polarization before and after transferring actually reflects a 180° distinction, which is also suggested by the phase-field simulation. LPFM scans for 30 u.c. BTO/Si shows consistent results with simulation predictions. The in-plane box-in-box pattern is switched, while the unbiased region sustains pristine high contrast multidomain structure. From the in-plane hysteresis loop in Fig. 2d, it is apparent that BTO membranes generate in-plane ferroelectricity after being released onto Si. Similar behaviors are also observed in other BTO/Si samples, with thicknesses varying from 4 u.c. to 160 u.c. (Supplementary Fig. 4). Notice that the actual switching voltage during scanning is dependent on the tip-surface contact, which is determined by both the deflection setpoints (Supplementary Fig. 5) and the tip conditions (Supplementary Fig. 6). And the switching voltage difference before and after transferring is mainly due to the absence of bottom electrode in BTO/SAO/STO and the presence of it in BTO/Si. Angle-resolved PFM measurements were carried out (Supplementary Fig. 7), revealing that in-plane polarization inside the randomly selected domain areas is mostly $a$ axis or $b$ axis equivalently oriented.

**Structural transition and the corresponding origin**. To study the polarization rotation after film releasing, systematic structural characterization has been performed by XRD on BTO films before and after transferring. By conducting high-resolution XRD $2\theta$-$\omega$ scans and reciprocal space mapping (RSM) measurements (Supplementary Fig. 8) for BTO samples of various thicknesses, we obtained lattice constants as shown in Fig. 3a. Compared with tetragonal phase in bulk BTO ($a = b = 3.992\,\text{Å}$, $c = 4.036\,\text{Å}$), the as-grown BTO films of all thicknesses show larger out-of-plane constants and smaller in-plane constants, which is resulted from the epitaxial compressive strain imposed by STO substrates. Sharing the same in-plane constant with STO substrates, the 15 u.c. as-grown BTO proved to be coherently strained. As the as-grown films become thicker, the relaxation effect becomes more prominent, leading to the monotonous increase of $a$ values and the decrease of $c$ values. In spite of this, for all the tested as-grown BTO, the $c$ is always apparently larger than bulk BTO while the $a$ is smaller, suggesting the stable $c$-axis tetragonal phase of the crystal structures.

The freestanding BTO membranes on Si display much different structural information from the as-grown ones. In Fig. 3a, freestanding BTO membranes of different thicknesses all present sharp reductions in out-of-plane lattice constant $c$ values compared with as-grown films, indicating general crystal structure collapses in c direction. In figure 3b, c, the RSM images around (002) and (103) diffraction peaks for BTO/Si suggest the structural composition of three different tetragonal phases (denoted by arrows in Fig. 3b, c). One of the origins of such structural variation is the relaxation of compressive strain after freestanding membranes are released from substrates. In addition, depolarizing field in freestanding membranes, which also tends to induce the formation of in-plane polarization[44–46] to preserve

polar order[52], can be another origin. Though compressive strain prevails over depolarizing field in BTO/SAO/STO, when the transferring process starts, depolarizing field also becomes influential in freestanding BTO as strain starts to be released. Therefore, through the transferring process, under the cooperative effects of strain relaxation and depolarization field, BTO membranes transform from $c$-axis tetragonal phase to a mixture of $a$-axis, $b$-axis, and $c$-axis tetragonal phases to stabilize ferroelectric polarization, as schematically illustrated in Supplementary Fig. 8c. It is also noteworthy that the peak intensities of $a$-axis and $b$-axis tetragonal phases in RSM images (Fig. 3b, c) are much stronger than that of $c$-axis phase, manifesting the dominance of in-plane tetragonal phases among the whole phase structures. Since each tetragonal phase ($a$ axis, $b$ axis, or $c$ axis) possesses polarization orientation along their elongated axis, the dominant form of $a$-axis and $b$ axis tetragonal phases thus accounts for the emergence of in-plane ferroelectricity throughout the freestanding BTO membrane, which is also consistent with our phase-field simulation result.

As structural transition occurred from pure $c$-axis phase to a combination of three tetragonal phases, the potential landscapes will be notably altered. The changes in potentials are schematically illustrated in Fig. 3d–g. In out-of-plane direction, the as-grown BTO layers prefer to polarize towards the STO substrates (called downward polarization), which may be attributed to the interface effect determined by the termination surface[48] (Fig. 3d). Nevertheless, during release, without clamping effect, the strong depolarizing field caused by initial out-of-plane polarization cannot be stabilized anymore, resulting in weakened $c$-axis ferroelectricity (Fig. 3e and Supplementary Fig. 3). Such depolarization, together with strain relaxation, prompts remarkable alteration in the potential profile of (001) plane and the generation of in-plane ferroelectricity. Compared to the as-grown BTO clamped on the substrate (Fig. 3f), freestanding BTO membrane possesses favorable in-plane polarization pointing along [100] and equivalent directions with minimum potential, as schematically shown in Fig. 3g. Such local in-plane ferroelectric instability comes from the formation of $a$-axis and $b$-axis tetragonal phases, implying the coexistence of polarization along $a$-axis and $b$-axis, which is in accordance with both the HADDF-STEM results (Supplementary Fig. 9) and our data from angle-resolved PFM measurements (Supplementary Fig. 7).

**Identification and manipulation of conductive DWs**. The pronounced in-plane ferroelectricity indicates BTO/Si an optimal platform for exploration and manipulation of the wanted conductive DWs. Using c-AFM (Fig. 4a) and transferring BTO on Au-coated Si, we identified the intrinsic conductive DWs in BTO/Si and further accomplished the controllable creation and elimination of conductive DWs through scanning probe techniques. Note that the in-plane multidomain structure of BTO/Si is formed the moment transferring process starts (before freestanding BTO touches the Au-coated Si), therefore the possible screening effect from Au doesn't affect the formation of these in-plane ferroelectric domains.

To gain further insight into the electrical transport characteristics of the prototype DW memory, current-voltage (I–V) curves collected on the DW (ON state) and of the DW (OFF state) are shown in Fig. 4b. The current signals observed here less than 0.5 nA at 2 V could be convoluted by the contact resistance, which can be improved if the measurement condition is optimized. The acquired I–V curve of ON state displays an onset of conductivity at 1.5 V, the necessary bias to annihilate the potential barriers between the top electrode (tip) and the conductive DWs[8,47]. Above 1.5 V, the current increases linearly

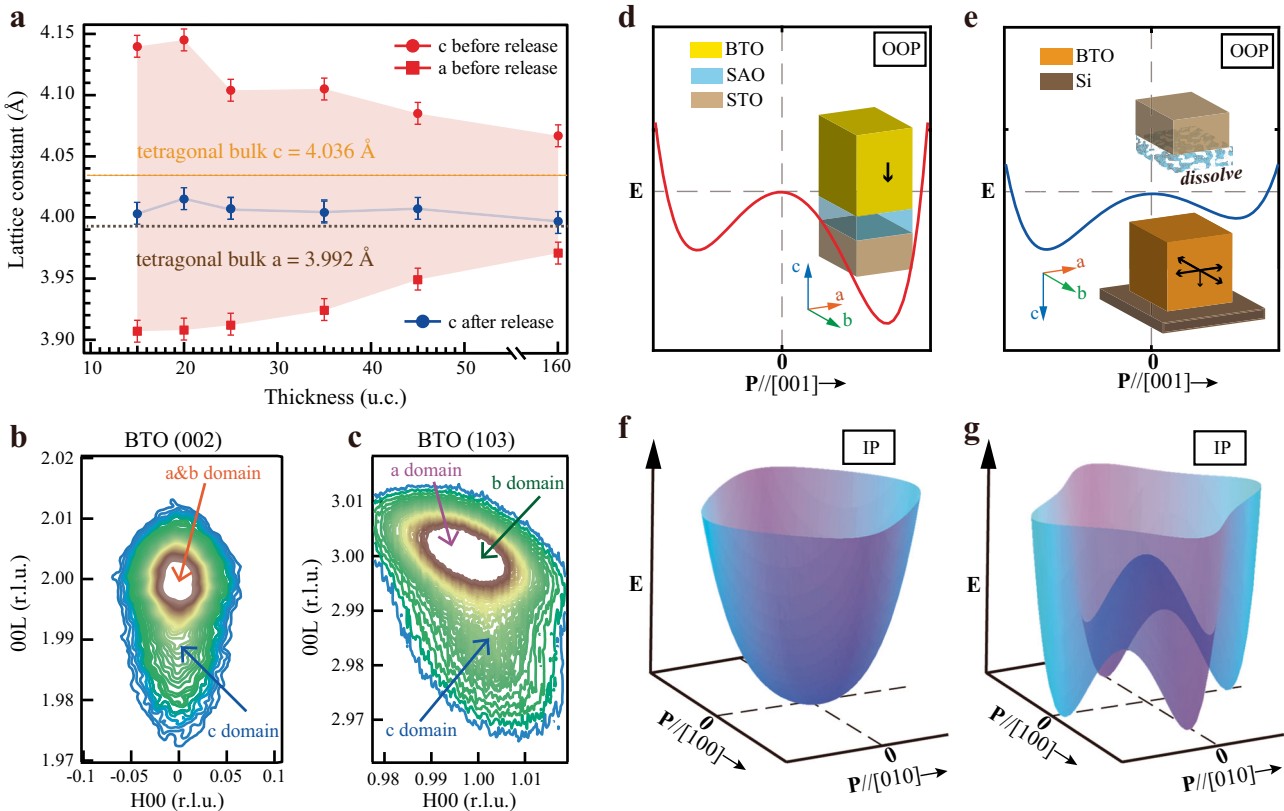

**Fig. 3 Structural transition of BTO. a** Lattice constants of BTO at different thicknesses before and after transferring. Bulk BTO lattice constants are offered for comparison in **a**. **b**, **c** RSM around (**b**) (002)- and **c** (103)-diffraction peaks. The mappings are based on a cubic structure with lattice constant of 0.4 nm. r.l.u., relative light units. Different parts of the peaks represent different domains, illustrated in detail by the arrows. Error bars are drawn considering instrument precision and accidental errors. **d**, **e** Qualitative potential diagram along [001] for BTO/SAO/STO in **d** and BTO/Si in **e**. Out-of-plane polarization directions are shown for both cases to clarify the *c*-axis polarization variation. **f**, **g** Qualitative in-plane potential diagrams for BTO/SAO/STO in **f** and BTO/Si in **g**.

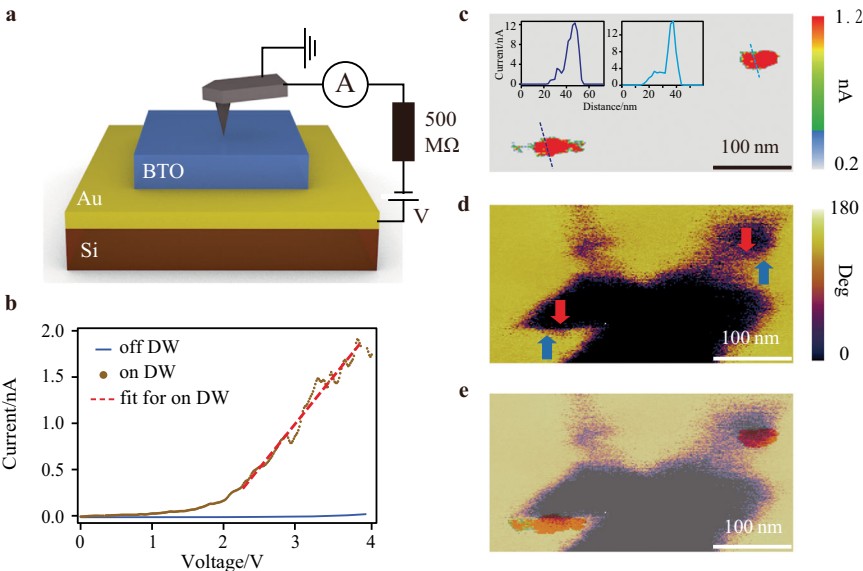

**Fig. 4 'Reading' of intrinsic conductive DWs. a** Schematics of c-AFM measurement circuit. **b** I–V curves on and off the conductive DWs. **c–e** Identification of the intrinsic conductive DWs. Current signals from c-AFM test (**c**) and in-plane phase signals from LPFM test (**d**) are overlapped for comparison (**e**). Current amplitudes along the line sections drawn in **c** are shown in insets of **c**. Arrows represent the polarization orientations in **d**.

along the voltage by fitting, which is a typical ohmic conduction behavior. Further, these I–V curves reveal a relatively high current contrast of about three orders of magnitude between OFF and ON states (ON/OFF = $10^3$) at a read bias of +4 V. Such ON/OFF ratio is around the same level of DW memories grown on perovskite substrates[10,18,19,21,22,27,53], signifying the comparable performance of our Si-based prototype as a candidate for non-volatile memories. These results were repeatable for randomly picked spots.

Figure 4c–e shows the identification of intrinsic conductive DWs in freestanding BTO, representing the 'reading' process of memory. In the demonstrated rectangular area, domain patterns were examined by LPFM while conductivities were tested by c-AFM. AFM images are confirmed to guarantee that measurements are taken in the same region. Polarization orientations for in-plane domains were verified by domain switching based on trailing field method[54,55] (Supplementary Fig. 10). Phase pattern in Fig. 4d displays distinct DWs, the regarding conductivities of which are demonstrated in c-AFM images (Fig. 4c) with representative current line profiles shown in insets. By overlapping two graphics, Fig. 4e reveals that only H-H DWs present notable conductivity (~nA), compared with those T-T DWs and domain regions (~pA). Similar selectivity has been previously reported in many other complex oxides[13,24,53,56], which manifests the n-type conductivity in the relaxed BTO membrane on Si, just like bulk BTO[8]. Free electrons as charge carriers may be provided by intrinsic excitation, biased tip injection, or charged defects such as oxygen vacancies[53]. Due to the band bending caused by positive bound charges, the electrons accumulate at H-H DWs, bringing out the detected conductivity. Additionally, since the band bending and charge accumulation strongly depend on the alignment of the DW with the polarization direction, only those H-H DWs nearly perpendicular to the polarization direction present prominent conductivity as depicted in Fig. 4e and further evidenced in Supplementary Fig. 11.

The ultimate step for memory functionality realization is to achieve artificial manipulation of conductive DWs through an external electric field, which represents the 'writing and erasing' process of memory. By scanning the tip over two regions with opposite biases (Supplementary Fig. 12d), we were able to create opposite polarization orientations in neighboring domain by the electric trailing field of the tip[54,55], and thus artificially create large-size H-H conductive DWs randomly over the membrane (Supplementary Fig. 12). Figure 5b, e, respectively demonstrate the domain structures before and after domain switching ('writing') on 15 u.c. BTO/Si. Evident H-H DWs appeared after switching, and were verified to be conductive in the later c-AFM measurement (Fig. 5f). When it comes to erasing the conductive DW, we can either turn it into an insulated T-T DW (Fig. 5h, m) or directly switch the whole region into a monodomain (Fig. 5k, n) to eliminate the DW. Both methods have proved to be effective in "turning off" the conductivity (Fig. 5i, l, o). Same writing and erasing process can also be applied to BTO/Si of different thicknesses (Supplementary Figs. 13, 14). The writing voltage for all tested samples ranges from 6 V to 8 V, which may be effectively lower by applying a closer tip-surface contact (Supplementary Fig. 5). And the reading voltage ranges from 3 V to 6 V. Moreover, the retention of the written conductive DW is also confirmed by a second 'reading' on the same sample after stored in air for eight months (Supplementary Fig. 14). The manipulation of the conductive DWs decisively displays the potential industrial applicability of perovskite/Si ferroelectric DW memories.

## Discussion

Apart from ON/OFF ratio, integration density is also a necessary concern in the application of non-volatile DW memories. From

the thickness-dependent LPFM results of freestanding BTO membranes, in-plane DW densities of our BTO/Si prototype are also analyzed. As shown in Supplementary Fig. 15, domain sizes tend to reduce as films become thinner, and thus the fractions of DWs in the whole material are also higher in thinner films. This relation between thickness and DW density suggests a potential path to improve the integration density of industrial DW memory. As the film thickness shrinks down, it is possible to obtain high-density conductive DWs as we elevate the number of DWs through dimensionality engineering.

The compatibility of BTO/Si with standard CMOS lithography processing, which makes this type of memory device possible, is also confirmed. In Supplementary Fig. 16, the LPFM results show similar strong in-plane ferroelectricity on a 20 u.c BTO membrane before and after an ion-beam lithography process, indicating the compatibility of BTO/Si with CMOS technology.

It is also worth mentioning that the widely reported ferroelectric DW memory with the architecture of coplanar metal electrodes[27,28,53,57] is also supposed to be conceptually applicable to our BTO/Si prototype, except that the previously reported coplanar-electrode models are usually designed in two-terminal fashion, where the two terminals are used for both 'writing' and 'reading', as illustrated in Fig. 1a. This two-terminal fashion may not be suitable to apply to our BTO/Si heterostructure since the electric field-created H-H DW is more likely to be perpendicular to the terminal-connecting direction. Therefore, a four-terminal fashion may be more appropriate for our heterostructure, where two terminals are used for 'writing' and the other two for 'reading' (Supplementary Fig. 17). Generally, for commercial electronic applications, the multi-terminal model is more practical but may be challenging for us currently, which will be explored in future investigations.

In summary, the in-plane ferroelectric multidomain structures have been realized and systematically revealed in BTO membranes after being transferred onto Si, the origin of which proved to be the collaboration of the strain relaxation and the depolarizing field-effect during release. With the in-plane ferroelectricity being the foundation, functionality of conductive DW memory was realized by the integrated prototype BTO/Si, as conductive DWs can be "read", "written", and "erased" in the heterostructure. The revealed 90° polarization rotation and our manipulation of conductive DWs in BTO/Si heterostructure offers a promising candidate for high-density non-volatile DW memories and provides a potential strategy for fabricating perovskite/Si alternatives to next-generation electronics including FETs and nanosensors.

## Methods

**Phase-field simulation**. The phase-field simulations are performed to calculate the ferroelectric polarization distributions in the BaTiO$_3$ membrane[58]. The time-dependent Ginzburg–Landau (TDGL) equation describes the temporal evolution of the polarization,

$$\frac{\partial P_i(r,t)}{\partial t} = -L \frac{\partial F}{\partial P_i(r,t)}, i = 1, 2, 3 \tag{1}$$

where $P_i(\boldsymbol{r}, t)$ is polarization, $\boldsymbol{r}$ is the spatial coordinate, $t$ is the evolution time, $L$ is the kinetic coefficient, and $F$ is the total free energy that includes the contributions from the Landau energy, the gradient energy, the elastic energy, and the electric energy:

$$F = \iiint_V (f_{Land} + f_{grad} + f_{elastic} + f_{ele}) \, dV \tag{2}$$

where $f_{Land}$, $f_{grad}$, $f_{elastic}$, and $f_{ele}$ denote the densities of the Landau energy, the gradient energy, the elastic energy density, and the electrostatic energy respectively. Electrostatic energy is where the depolarizing field is taken into consideration. The mathematical expressions for these energy densities of the BTO membrane are described in the literature[59]. The thickness of the BTO membrane is 10 nm, and the applied misfit strain due to the different lattice constants of the BTO membrane and STO substrate are obtained from the literature[60]. For BTO thin film on STO

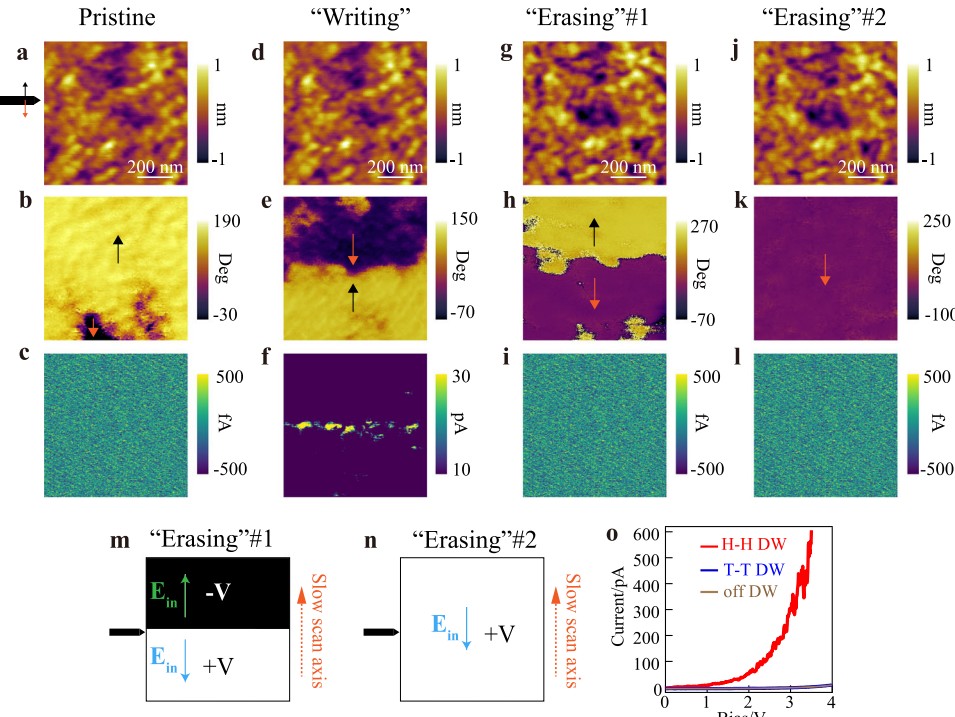

**Fig. 5 Writing and erasing conductive DWs in 15 u.c. BTO/Si. a–c** Pristine state of the tested region. From top to bottom: AFM height image, LPFM phase image, c-AFM current image (same for **d–l**). **d–f** Results after 'writing' process. **g–i** Results after 'erasing' process which switch the H-H DW into T-T DW, scanning method shown in **m**. **j–l** Results after 'erasing' process which switch the whole region into monodomain, scanning method shown in **n**. **o** I–V curves measured at different positions.

substrate, a built-in electric bias ~0.001 V is added. The parameters in the simulation are listed in Supplementary Table 1.

**Film growth**. Epitaxial SAO buffer layers and BTO films were grown on (001) $TiO_2$-terminated single-crystalline STO substrates in sequence by oxide molecular beam epitaxy. SAO layers were first deposited at a substrate temperature of 850 °C with an oxygen pressure of $1 \times 10^{-6}$ Torr. BTO layers were subsequently deposited at 850 °C in an oxidant background of $1 \times 10^{-6}$ Torr (10% ozone and 90% oxygen). During the growth, RHEED was employed to monitor the surface quality and precisely control the thicknesses of the films.

**Film transfer**. As-grown BTO samples were transferred onto doped Si by applying a homemade clamping device to obtain atomically smooth surfaces of BTO membranes (Supplementary Fig. 1). Two groups of screw holes were made on each plate, with screws fixed on the bottom plate through one group beforehand. For c-AFM measurements, gold was deposited on Si by electron beam lithography before transferring. One 10 mm × 10 mm Si substrate was laid on the bottom plate, after which the BTO/SAO/STO sample was placed upside down onto Si so that the BTO side was in contact with the Si base. The top plate was mounted at the end by tightening the screws through the other group of holes so that the as-grown BTO films and Si substrate were clamped during the transferring process. The whole apparatus was immersed in de-ionized water for 12 hours to dissolve the SAO layer and then taken out. After dried by $N_2$ flow, the BTO/Si heterostructure was ultrasonically cleaned by acetone, isopropanol, and de-ionized water in series, with 5–10 minutes for each cleaning.

**AFM and PFM measurements**. Both film surface micromorphology and film ferroelectricity were characterized by Asylum Research MFP-3D Origin+ scanning probe microscopy (SPM). Out-of-plane and in-plane ferroelectric polarization were tested by VPFM and LPFM respectively. For PFM measurements, Dual AC resonance tracking (DART) mode was adopted, with ac bias of 2 V applied to Pt/Ir coating Si scanning tips during reading. As for the domain writing process, positive or negative biases (amplitudes vary by film thickness) were applied on the tip when scanning different regions, so as to create designed domain patterns.

**XRD measurements**. BTO crystalline structures were characterized by high-resolution XRD with a Bruker D8 Discover diffractometer. $2\theta$-$\omega$ scanning and RSM test were both performed to confirm the high crystallinity and extract the lattice constants of BTO samples.

**c-AFM measurements**. The same SPM instrument and tips for PFM were used for c-AFM measurements. Instead of making tip biased like PFM, the tip in c-AFM mode is treated as the virtual ground while applying 3–6 V bias voltage on the supporting Au-coated Si for current mappings. LPFM phase images before and after c-AFM measurements have been tested to ensure conductive tips are non-destructive to existing domain structures. For c-AFM mode, the current was measured directly in the contact mode through a built-in current amplifier (0.5 pA to 20 nA) with a protect resistance of 500 MΩ. In order to weaken the influence of charging and discharging behavior when measuring I–V curves (Supplementary Fig. 18), I–V data can be obtained with a lower sweep frequency at only the positive bias region.

## Data availability

All data needed to evaluate the conclusions in the paper are present in the paper and/or the Supplementary Information. Additional data related to this paper may be requested from the authors.

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

## Acknowledgements
We thank Xiaoqing Pan, Di Wu, Darrell G. Schlom, Yueying Li, and Wenjie Sun for fruitful discussions. This work was supported by the National Key R&D Program of China (Grant No. 2021YFA1400400 and 2018YFA0305800), the National Natural Science Foundation of China (Grant Nos. 11861161004, 51972028, and 51772145) and the Fundamental Research Funds for the Central Universities (Grant no. 0213-14380198).

## Author contributions
Y.F.N. conceived and directed the experiments. Assisted by T.Y.G. and J.H.G., H.Y.S. grew the BTO/SAO/STO heterostructures. Using clamping device made by J.F.Y., H.Y.S., J.R.W., Y.W.L. and T.J.Z. integrated the prototype through transferring. C.Q.G. and H.B.H. made the phase-field simulation. Y.S.W. and Y.F.H. completed electrode deposition by electron beam lithography. H.Y.S. and J.R.W. performed the ferroelectricity characterization and analyzed the PFM data with the help of H.Y.F. J.R.W. and J.H.G. conducted XRD measurements with H.Y.S. analyzing the data. J.R.W. and H.Y.S. completed c-AFM tests with the help of Y.W.L. and T.T.Z. W.M. and P.W. conducted STEM measurements. Y.F.N., J.R.W. and H.Y.S. wrote the manuscript. All authors discussed the data and contributed to the manuscript.

## Competing interests
The authors declare no competing interests.
