## [Peer Review File · Nature Communications]

Nonvolatile ferroelectric domain wall memory integrated on siliconREVIEWER COMMENTS

Reviewer #1 (Remarks to the Author):

In this work, the authors demonstrated a domain wall memory prototype that is made up of freestanding BaTiO₃ membranes that have been transplanted to silicon. Even though BTO films grown on (001) SrTiO₃ substrates as-grown are completely c-axis polarized, the authors found that when they are released from the substrate and integrated onto silicon, they exhibit distinct in-plane multidomain structures, which we attribute to the combined effects of the depolarizing field and strain relaxation. Although it is interesting to observe the in-plane effect in BTO membranes, there are insufficient information and not enough evidence of how in-plane ferroelectricity are induced and the Si substrate effect. Additional control measurements and quantitative analyzes mentioned below in details are needed. When the authors address these issues, I would like to reconsider the publication of this work.

“The authors show successful demonstration of transferred BTO on a silicon substrate. However, I doubt that the effect of bottom substrate (Si) is well measured. As the Si substrate is interfaced with a BTO film by wet transfer in this case, there would be two possible scenarios: (1) BTO film is conformally (well) transferred on the Si substrate, (2) BTO film does not make a good interface with Si. For (1) case, I would like to further understand the coupling effect of Si substrate and how this can be integrated with CMOS. Also, the authors may want to confirm whether (2) case happens in the experiment, which could a huge bottleneck for application of BTO freestanding films.”

“The authors argue that reading and writing operations were performed in conductive DWs at Figure 4. However, it is challenging to claim that domain switching happens based on the results of Figure 4(f) and Figure 4(g). If the thickness of conductive DWs is at nanoscale, the conductive DWs can be damaged by a cantilever tip, which can create current path. I would suggest to conduct experiments on diverse thickness of BTO films.”

“If I understood correctly, the conductive DWs can be created artificially here. In this case, the authors need to show how they can control DWs’ programming. I would like to know whether the authors can erase the programmed DWs.”

“It would be quite intriguing if the authors are able to fabricate simple ferroelectric devices. I believe that one of advantages of this structure is the capabilities of hetero-integration with silicon substrate which can bring advanced CMOS technologies.”

Reviewer #2 (Remarks to the Author):

Haoying Sun et al. reported transferred BaTiO₃ membranes onto Si wafer with ferroelectric polarization rotated from out-of-plane to in-plane configuration, which is verified by vector PFM. Overall, I found the data is supportive of the H-t-H conductive domain wall scenario. However, I am not convinced these conductive ferroelectric domain walls can be useful for memory technology, which the authors have emphasized many times in the abstract and introduction. Please find my concerns/questions as follows:

1. The challenges to implementing this type of in-plane domain walls are 1) in reality (also

shown in figure 1a), it is almost impossible to create an H-t-H domain wall consistently with a two-terminal device fashion. A scanning probe approach is of course possible since one can precisely control the in-plane field by changing the slow scan axis. Can the author demonstrate a working single device instead of scanning probe-based demonstration? 2) The switching voltage is too high for practical consideration. 3) thermal stability (retention) is a concern for charged domain walls. Can the authors provide some retention data, which is crucial for memory technology, to support your statement? 4) CAFM is a two-terminal measurement that is highly contact-dependent especially for the resistive materials (in this case less than 0.5 nA at 2 volts). The behaviors observed here could be convoluted by the contact resistance.

2. In figure 2a, the out-of-plane amplitude does not look correct. For a reversed out-of-plane box-in-box switching, the amplitude channel should only have contrast at the domain walls.

3. In figure 2, why the switching voltages are so different before and after transferring?

4. After transferring, from the c/a ratio the BTO seems to become a cubic phase (symmetry). Can the authors explain why ferroelectricity is still preserved? High-resolution RSMs shall be able to tell the in-plane tetragonal domains.

5. The authors mentioned the lack of screening results in a large depolarization field. I am wondering did not the author use Au/Si wafer as the bottom electrode for CAFM measurements? There should be enough free electrons for screening from the Au. Can the author comment on it?

6. In figure 4b, the blue line is denoted as "off DW" showing a constant current level at even zero bias. The current level on DW is even lower at zero bias. Can the author comment on it?

7. In figure 4c and 4i, not all the H-t-H walls are showing conductive behaviors. For example, at the bottom right of figure 4c, there should be also an H-t-H domain wall according to the author's remarks, but it seems not conductive or at least not as conductive as the marked ones. Can the author comment on it?

8. In figure S1, it does not look like 6 u.c. deposition of SAO to me.

In conclusion, this paper reported the head-to-head conducting domain walls of BaTiO₃ films on Si, which is not surprising for the ferroelectrics. Some technical details can be improved or clarified. But I found it is hard to convince me this article has significant breakthrough from the previous studies (Science advances 3, no. 6 (2017): e1700512., Nature nanotechnology 13, no. 10 (2018): 947-952., Nature nanotechnology 10, no. 2 (2015): 145-150., and Advanced Materials, 32(9), p.1905132.). Therefore I do not recommend publishing this research article before revision.

Reviewer #3 (Remarks to the Author):

The authors report on the fabrication and switching of freestanding ferroelectric BaTiO₃ (BTO) membrane integrated on a Si wafer. By releasing the epitaxial tetragonal BTO film using the Sr₃Al₂O₆ buffer layer, the initially c-axis polarized film transforms into a mixture of a-, b-, and c-axis polarized domains, exhibiting an apparent cubic structure. The released

BTO membrane therefore show in-plane polarization which can be switched via vertically applied bias functioning as a memory element.

Freestanding single crystalline oxide membranes are one of the growing areas of research not only from academic but also from industrial perspectives. Ferroelectrics, which is one of the major properties of complex oxides, is especially an attractive target for membrane applications.

Overall, this paper is well organized but I have a few questions regarding the experiments and interpretations, which I would like the authors to address before considering further.

1. In Figs. 4f and 4g, the authors show domain switching in a very small area. Do the authors have larger area maps before/after switching? Given the emphasis on application, I think the actual scale of the membrane as well as the polarization map would be very instructive.

2. Also, I am concerned about the concentrated electric field lines close to the c-AFM tip when 6V is applied on the entire back side of the membrane. The area ratio must be gigantic given the small radius of curvature of the AFM tip. Consequently, wouldn't a large electric be applied locally around the tip, which readily switches the domains and leave parts of the membrane far from the tip unswitched?

3. How were the direction of the polarization determined in Fig. 4d, 4f, and 4g, i.e. the red and blue arrows? Also, in Figs. 4e, 4i, the overlapping of the current map to the LPFM map are not convincing. The authors should measure areas with markers active for both maps to identify that the measurements were taken on the same area.

4. In Fig. 4b, the current observed for on-DW shows a few nA. However, in the current maps of Fig. 4e, 4i, the scale bar shows up to 160 pA, an order of magnitude smaller than expected at 4V. I would like the authors to first show statistics and secondly to explain if there is possibility of amorphous or non-ferroelectric insulating regions acting as additional resistance. BTO is known to be hydroscopic and even though their film may be single crystalline, 12h immersion in water may selectively dissolve the surface Ba ions, leaving TiO₂ or Ba-deficient BTO in the membrane.

5. By switching the polarization, can the authors show that the initially H-H DW turns into T-T DW by taking I-V cures before/after switching?

6. The authors show a schematic of the potential landscape in Fig. 3c-f. The x-axis, which I suspect is the polarization along [001] or [010], does not have an origin. Therefore, the asymmetry in Fig. 3c is not obvious. I assume the dotted vertical line is $P = 0$, but the authors should clarify.

7. LL 114-117. Do the authors have evidence to support this claim? Is Fig. 2a vs. 2c? I would like the authors to be more quantitative in their description.

Responses to Reviewer #1:

In this work, the authors demonstrated a domain wall memory prototype that is made up of freestanding BaTiO₃ membranes that have been transplanted to silicon. Even though BTO films grown on (001) SrTiO₃ substrates as-grown are completely c-axis polarized, the authors found that when they are released from the substrate and integrated onto silicon, they exhibit distinct in-plane multidomain structures, which we attribute to the combined effects of the depolarizing field and strain relaxation. Although it is interesting to observe the in-plane effect in BTO membranes, there are insufficient information and not enough evidence of how in-plane ferroelectricity are induced and the Si substrate effect. Additional control measurements and quantitative analyzes mentioned below in details are needed. When the authors address these issues, I would like to reconsider the publication of this work.

We thank Referee #1 for the positive review and constructive questions/suggestions. As suggested by the Referee, we have performed phase-field simulations and additional experiments, which support our observations and conclusions. More details can be seen in the following point-by-point responses to Referee #1's questions and concerns.

Q1: There are insufficient information and not enough evidence of how in-plane ferroelectricity are induced and the Si substrate effect.

AI: We thank the Referee for pointing out this issue. The in-plane ferroelectricity in freestanding BTO membranes is formed as a result of the combined effects of strain relaxation and depolarizing field during transferring process. Note that the strain relaxation should take place when BTO is no longer bonded with the STO substrate during the dissolution of the sacrificial buffer layer. Under the effect of strain relaxation together with the depolarizing field in BTO membranes, the polarization rotation takes place before the freestanding films attached onto the silicon wafer. As such, the Si substrate does not play an important role in forming the in-plane ferroelectricity in BTO membranes.

To confirm the above scenario, we have also conducted phase-field simulations on BTO freestanding films (Fig. A1) by taking the strain relaxation and depolarizing field effects into account. Indeed, the results show a clear in-plane ferroelectricity, which agree well with our experimental observations.

Moreover, in addition to the XRD and PFM experiments showing the in-plane ferroelectricity in BTO membranes, we further carried out atomic-scale polarization mapping based on the plan-view high-angle annular dark-field scanning transmission electron microscopy (HAADF-STEM) measurements as shown below (Supplementary Fig. 9). Prominent *a*-axis and *b*-axis ferroelectric domains exist in freestanding BTO membranes, which is consistent with our XRD and PFM observations.

We have revised our manuscript and Supplementary Information accordingly.

In the main text, we have added phase-field simulation results as Fig. 1b and the corresponding explanation in Page 4 (text in blue): ‘**Anticipation for in-plane ferroelectricity.** The polarization distribution of 25 u.c. BTO thin films before and after releasing from the STO substrate are calculated by phase-field simulations (Fig. 1b). The effects from depolarizing field, epitaxial strain (clamped film) and strain relaxation (freestanding membranes) are considered (see Methods). The BTO thin film clamped on STO substrate demonstrates monodomain with *c*-axis downward polarization, in agreement with previous reports [Ref 48]. However, when the epitaxial strain induced by lattice mismatch vanishes along with the separation from STO substrate, the freestanding BTO membrane exhibits prominent in-plane multidomain structure. Such in-plane ferroelectricity can be a fertile ground for conductive DWs, encouraging us to experimentally apply freestanding BTO/Si system as a potential prototype for DW memories.’

In the main text, Page 8 (text in blue): ‘Since each tetragonal phase (*a*-axis, *b*-axis or *c*-axis) possesses polarization orientation along their elongated axis, the dominant formation of *a*-axis and *b*-axis tetragonal phases thus accounts for the emergence of in-plane ferroelectricity throughout the freestanding BTO membrane, which is also consistent with our phase-field simulation result.’

In the Supplementary Information, we have added HADDF-STEM images as Supplementary Fig. 9.

Figure A1 | Phase-field simulation of BTO thin films. **a**, Simulated domain structure of as-grown 25 u.c. BTO thin film fully strained on the STO substrate showing pure downward (point to STO substrate) out-of-plane polarization. **b**, Simulated domain structure of a fully relaxed 25 u.c. freestanding BTO membrane showing prominent in-plane domain structures.

Figure S9 | In-plane ferroelectricity of BTO membrane. **a,b**, Maps of polar atomic displacement vectors obtained from the plan-view HAADF-STEM images of a 160 u.c. BTO membrane transferred to a holey carbon TEM grid. **c,d**, Zoom-in atomically resolved HAADF-STEM images. The average displacement of Ti atom is about 21 pm for ferroelectric domain in **a** and 17 pm for domain in **b**.

[48] Guo, R, *et al.* Tailoring Self-Polarization of BaTiO₃ Thin Films by Interface Engineering and Flexoelectric Effect. *Adv. Mater. Interfaces* **3**, 1600737 (2016).

Q2: The authors show successful demonstration of transferred BTO on a silicon substrate. However, I doubt that the effect of bottom substrate (Si) is well measured. As the Si substrate is interfaced with a BTO film by wet transfer in this case, there would be two possible scenarios: (1) BTO film is conformally (well) transferred on the Si substrate, (2) BTO film does not make a good interface with Si. For (1) case, I would like to further understand the coupling effect of Si substrate and how this can be integrated with CMOS. Also, the authors may want to confirm whether (2) case happens in the experiment, which could a huge bottleneck for application of BTO freestanding films.

A2: We thank the Referee for raising the important question about the interface between BTO and Si. To address this question, we have performed microscopic structural characterizations on our samples. As shown below (Supplementary Fig. 2), the cross-sectional low-magnification HAADF-STEM and Energy-dispersive X-ray spectroscopy (EDS) images of BTO/Si sample show a straight and sharp interface between BTO membrane and Si, indicating that BTO membrane is in good contact with silicon.

To confirm the compatibility of BTO/Si with CMOS technology, following Referee's advice, we further conducted standard ion-beam lithography on BTO/Si (Detailed procedure in Supplementary Fig. 15). Before and after lithography process, the freestanding BTO membrane exhibit similar strong in-plane ferroelectricity, indicating BTO/Si can be compatible with CMOS technology.

We have also revised our manuscript and Supplementary Information based on the results of the above measurements.

In the main text, Page 5 (text in blue): ‘As indicated by high-angle annular dark-field scanning transmission electron microscopy (HAADF-STEM) and energy-dispersive X-ray spectroscopy (EDS) measurements (Supplementary Fig. 2), the BTO membrane conformally attaches to Si substrate.’

In the main text, Page 12 (text in blue): ‘The compatibility of BTO/Si with standard CMOS lithography processing, which makes new type of memory devices possible, is also confirmed. In Supplementary Fig. 15, the LPFM results show similar strong in-plane ferroelectricity on a 20 u.c BTO membrane before and after an ion-beam lithography process, indicating the compatibility of BTO/Si with CMOS technology.’

In the Supplementary Information, we have shown HAADF-STEM and EDS images of BTO/Si interface as Supplementary Fig. 2.

In the Supplementary Information, we have displayed LPFM results of BTO on Si before and after ion-beam lithography as Supplementary Fig. 15.

Figure S2 | HAADF-STEM and EDS measurements of a BTO/Si sample. a, HAADF-STEM image of Pt/C(conductive protection layer)/BTO/Si cross-sectional sample, revealing a good contact between BTO membrane and Si substrate. **b-g**, Corresponding EDS elemental maps showing the spatial distribution of Ba, Ti, O, Si, C and Pt. Note that the distribution of O in Si substrate is due to the oxidization during the TEM sample preparation and storage.

Figure S15 | LPFM results before and after ion-beam lithography on 20 u.c. BTO/Si. a, Schematic illustration for fabricating square-patterned BTO on Si, through electron-beam lithography (EBL) and Ar⁺ ion-beam lithography (IBE). **b-d,** LPFM results of pristine BTO/Si. **e,** LPFM results of BTO/Si after ion-beam lithography.

Q3: The authors argue that reading and writing operations were performed in conductive DWs at Figure 4. However, it is challenging to claim that domain switching happens based on the results of Figure 4(f) and Figure 4(g). If the thickness of conductive DWs is at nanoscale, the conductive DWs can be damaged by a cantilever tip, which can create current path. I would suggest to conduct experiments on diverse thickness of BTO films.

A3: We fully understand the Referee's concern that conductive paths may be created due to sample damages by the cantilever tip. To address that, we have conducted AFM and LPFM measurements before and after c-AFM experiments. As shown in Supplementary Fig. 13, the surface topography (first row) and domain structure (second row) both maintain their original states after c-AFM test (and also after 8 months). Moreover, the conductive DWs can be successfully erased by domain switching in two different ways (details in A4, Fig. 5 and Supplementary Fig. 13), which manifests that the conductive paths are created as a result of domain switching instead of the sample destruction.

Following the Referee's suggestion, as shown in Fig. 5, we also performed similar writing and reading on sample with different thickness (15 u.c. BTO/Si), which reproduces the same result as that in the 20 u.c. sample. We have also run the LPFM measurements on the sample before and after

c-AFM measurements, which show no impact on the domain structure in the sample (Fig. A2).

We have revised our manuscript as well as Supplementary Information to provide information about the above supportive measurements.

In the main text, main results of artificial ‘writing, reading and erasing’ have been displayed in Fig. 5. Relevant illustration is added in Page 11 (text in blue): ‘Fig. 5b and Fig. 5e respectively demonstrate the domain structures before and after domain switching (‘writing’) on 15 u.c. BTO/Si. Evident H-H DWs appeared after switching, and was verified to be conductive in the later c-AFM measurement (Fig. 5f). When it comes to erasing the conductive DW, we can either turn it into an insulated T-T DW (Fig. 5h, m) or directly switch the whole region into a monodomain (Fig. 5k, n) to eliminate the DW. Both methods have proved to be effective in ‘turning off’ the conductivity (Fig. 5i, l, o). Same writing and erasing process can also be applied to BTO/Si of different thickness (Supplementary Fig. 12, 13).’

In the main text, Page 16 (text in blue): ‘LPFM phase images before and after c-AFM measurements have been tested to ensure conductive tips being nondestructive to existed domain structures.’

In the Supplementary Information, we have provided detailed ‘writing’ results on 20 u.c. BTO/Si in Supplementary Fig. 12.

In the Supplementary Information, we have demonstrated the AFM and LPFM phase images of 20 u.c. BTO/Si in the original version, before the 8-months-later c-AFM measurement and after the 8-months later c-AFM measurement in Supplementary Fig. 13 (first two rows).

Figure S13 | Retention and erasure of the conductive DWs on 20 u.c. BTO/Si. a, The written conductive DWs on 20 u.c. BTO/Si. **b,** Scanning of the created conductive DWs 8 months after creating it. Reading voltage is at +3 V. **c,** Erasing the created conductive DW by scanning over the whole region with the same negative voltage. From top to bottom: height image, LPFM phase image (arrows for polarization directions), c-AFM current image and phase-current overlapping image.

Figure 5 | Writing and erasing conductive DWs in 15 u.c. BTO/Si. **a-c**, Pristine state of the tested region. From top to bottom: AFM height image, LPFM phase image, c-AFM current image (same for **d-l**). **d-f**, Results after ‘writing’ process. **g-i**, Results after ‘erasing’ process which switches the H-H DW into T-T DW, scanning method shown in **m**. **j-l**, Results after ‘erasing’ process which switches the whole region into monodomain, scanning method shown in **n**. **o**, I-V curves measured at different positions.

Figure A2 | Domain patterns before and after c-AFM measurement on 15 u.c. BTO/Si. **a**, Phase image before c-AFM scanning. **b**, c-AFM current image. **c**, Phase image before c-AFM scanning.

Figure S12 | ‘Writing’ and ‘reading’ H-H DW on 20 u.c. BTO/Si. a,b, AFM height image (a) and LPFM phase image (b) before ‘writing’. **c,d,** AFM height image (c) and LPFM phase image (d) after ‘writing’. **e,f,** AFM height image (e) and c-AFM current image (f) by ‘reading’ process.

Q4: If I understood correctly, the conductive DWs can be created artificially here. In this case, the authors need to show how they can control DWs’ programming. I would like to know whether the authors can erase the programmed DWs.

A4: As mentioned in **A3**, following this suggestion, we have carried out the erasure of conductive DWs on 15 u.c. and 20 u.c. BTO/Si samples. One way to erase the conductive DW is to simply apply the same bias to the whole region to turn it into in-plane monodomain (Fig. 5, Supplementary Fig. 13). Another way, as suggested by Referee #3, is to write the head-to-head (H-H) DW into a tail-to-tail (T-T) DW, which also shows no conductivity (Fig. 5).

We have revised our manuscript and Supplementary Information for completeness based on these results (demonstrated above in **A3**).

Q5: It would be quite intriguing if the authors are able to fabricate simple ferroelectric devices. I believe that one of advantages of this structure is the capabilities of hetero-integration with silicon substrate which can bring advanced CMOS technologies.

A5: We thank the Referee for pointing out the significance of device fabrication, which we can’t agree more. As discussed in **Q2**, following Referee’s advice, we have operated ion-beam lithography on BTO/Si heterostructure. The preserved ferroelectricity in freestanding BTO after processing shows strong compatibility of BTO/Si with CMOS technology, manifesting the applicability of BTO/Si for industrial device fabrication. In order to make a realistic device, terminals are more favored than probing tip for manipulation of conductive DWs. However, the multi-terminal device

requires high precision of nanofabrication beyond our capability at this point, but this is exactly where we are heading to in the next step.

We have also added such discussion into our main text, Page 13 (text in blue): ‘Generally, for commercial electronic applications, multi-terminal model is more practical but maybe challenging for us currently, which will be explored in future investigations.’

Responses to Reviewer #2:

Haoying Sun et al. reported transferred BaTiO₃ membranes onto Si wafer with ferroelectric polarization rotated from out-of-plane to in-plane configuration, which is verified by vector PFM. Overall, I found the data is supportive of the H-t-H conductive domain wall scenario. However, I am not convinced these conductive ferroelectric domain walls can be useful for memory technology, which the authors have emphasized many times in the abstract and introduction.

We thank the Referee for the positive review and the constructive comments, especially the concern about device applications. We have answered the questions in detail in the following pages and have revised our manuscript accordingly.

*Q1.1: The challenges to implementing this type of in-plane domain walls are **I**) in reality (also shown in figure 1a), it is almost impossible to create an H-t-H domain wall consistently with a two-terminal device fashion. A scanning probe approach is of course possible since one can precisely control the in-plane field by changing the slow scan axis. Can the author demonstrate a working single device instead of scanning probe-based demonstration?*

A1.1: We thank the Referee for raising the concern of two-terminal device. We fully agree with the Referee that it is a challenge to consistently create conductive H-H DW in the two-terminal (coplanar terminals) device, since the formation of H-H DW between two coplanar electrodes is more likely to be perpendicular to the terminal-connecting direction.

As BTO/Si requires relatively strict H-H DW, what may be plausible for BTO system is the four-terminal fashion (Supplementary Fig. 16) where two terminals are used to write or erase the DW while the other two are used for ‘reading’ the current. But this four-terminal fashion not only requires high fabrication precision, but also needs to promise well ohmic contact between electrode and conductive DWs, which is so far beyond our nanofabrication capability. Hopefully we will be able to achieve coplanar four-terminal fashion in our future investigation.

The main purpose of our work is to realize in-plane ferroelectricity and conductive DWs in freestanding perovskite BTO membranes and integrate them to Si to demonstrate the possibility of a DW memory prototype. As directly growing perovskite oxides on Si substrates has long been hindered by issues like interface oxidation and large lattice mismatch, only a very limited number of perovskite oxides such as SrTiO₃ have been demonstrated to be able to synthesized on silicon using a very sophisticated and time-consuming method [Ref 34]. Here, our work makes a breakthrough in integrating high-quality crystalline ferroelectric BaTiO₃ membranes on silicon using a growth and transfer method. Moreover, a very interesting polarization switching takes place in freestanding BTO films driven by a combination of strain-relaxation and depolarization field, which agrees with our phase-field simulation. We further demonstrate the realization of rewritable conductive DWs in this heterostructure, providing preliminary basis for future device applications.

Although two-terminal model would be challenging for BTO, previous reports [Ref 27, 28, 53, 57] have suggested that it is applicable to other perovskite systems especially BiFeO₃ (BFO). Nonetheless, our work provides the first example of the integration of conductive DWs in ferroelectric perovskite oxides on silicon, inspiring the community to make more systematic studies in integrating the rich functionalities of perovskite oxides with silicon for industrial electronic applications in the near future.

Following Referee's advice, we have revised our manuscript and Supplementary Information to discuss the limitation and potential of applying two- or multi-terminal model onto BTO/Si prototype.

In the main text, we have discussed the limitation and potential of our BTO/Si on Page 12 (text in blue): 'It is also worth mentioning that the widely reported ferroelectric DW memory with the architecture of coplanar metal electrodes [Ref 27, 28, 53, 57] is also supposed to be conceptually applicable to our BTO/Si prototype, except that the previously reported coplanar-electrode models are usually designed in two-terminal fashion, where the two terminals are used for both 'writing' and 'reading', as illustrated in Fig. 1a. This two-terminal fashion may not be suitable to applied to our BTO/Si heterostructure since the electric field-created H-H DW is more likely to be perpendicular to the terminal-connecting direction. Therefore, a four-terminal fashion may be more appropriate for our heterostructure, where two terminals are used for 'writing' and the other two for 'reading' (Supplementary Fig. 16). Generally, for commercial electronic applications, multi-terminal model is more practical but maybe challenging for us currently, which will be explored in future investigations.'

In the Supplementary Information, the schematics of four-terminal design is displayed in Supplementary Fig. 16.

Figure S16 | Schematics of a four-terminal model. Four-terminal model may be applicable to BTO/Si prototype, with two terminals (purple) used for 'writing' and the other two (brown) for 'reading'. Arrows represent polarization orientations.

[27] Jiang, J, *et al.* Temporary formation of highly conducting domain walls for non-destructive read-out of ferroelectric domain-wall resistance switching memories. *Nat. Mater.* **17**, 49-56

(2018).

[28] Sharma, P, *et al.* Nonvolatile ferroelectric domain wall memory. *Sci. Adv.* **3**, e1700512 (2017).

[34] Warusawithana, MP, *et al.* A Ferroelectric Oxide Made Directly on Silicon. *Science* **324**, 367 (2009).

[53] Sharma, P, *et al.* Conformational Domain Wall Switch. *Adv. Funct. Mater.* **29**, 1807523 (2019).

[57] Yang, W, *et al.* Nonvolatile Ferroelectric-Domain-Wall Memory Embedded in a Complex Topological Domain Structure. *Adv. Mater.* **34**, e2107711 (2022).

Q1.2: 2) *The switching voltage is too high for practical consideration.*

A1.2: We fully agree with the Referee that the switching voltage in our current experiments is so far too high for practical device applications. The reason for that is most likely related to the exact scanning conditions we used for the domain switching. The electric field effectively applied on the film is rather low since there is always a gap (e.g. contact resistance) between the tip and the sample. As shown below (Supplementary Fig. 5), by applying larger force to obtain a better contact between the tip and BTO surface (increasing the deflection setpoint), we can greatly lower switching voltage from ~7 V to ~4 V, which is similar or lower than the values reported in the epitaxial ferroelectric films in literatures (6 V [Ref 21], 8 V [Ref 28], 5-16 V [Ref 29]).

As also suggested by the Referee, multi-terminal pattern should be used for industrial applications. With better ohmic contact and pattern design, lower switching voltage may be achievable by fabricating terminals. Nonetheless, the main contribution of this work is the observation of in-plane ferroelectricity in relaxed ultrathin BTO membranes and the successful integration of conductive ferroelectric DWs on silicon, with application details to be optimized in future systematic investigations.

We have revised our manuscript and Supplementary Information to include the above discussion.

In the main text, Page 7 (text in blue): ‘Notice that the actual switching voltage during scanning is dependent on the tip-surface contact, which is determined by both the deflection setpoints (Supplementary Fig. 5) and the tip conditions (Supplementary Fig. 6).’

In the main text, Page 11 (text in blue): ‘The writing voltage for all tested samples ranges from 6 V to 8 V, which may be effectively lower by applying a closer tip-surface contact (Supplementary Fig. 5).’

In the Supplementary Information, we have demonstrated amplitude loops at different deflection setpoints and domain switching results (by trailing field [Ref 54, 55]) with lower switching voltage in Supplementary Fig. 5.

Figure S5 | Lower switching voltage by higher deflection setpoints. **a**, Butterfly-shape amplitude loops at different deflection setpoints. Coercive fields (voltages) drop with higher deflection setpoints (better tip-surface contact). **b**, Trailing field method for domain switching. **c,d**, Domain switching results at 4 V by applying higher deflection setpoint.

- [21] Li, L, *et al.* Giant Resistive Switching via Control of Ferroelectric Charged Domain Walls. *Adv. Mater.* **28**, 6574-6580 (2016).
- [28] Sharma, P, *et al.* Nonvolatile ferroelectric domain wall memory. *Sci. Adv.* **3**, e1700512 (2017).
- [29] Jiang, AQ, *et al.* Ferroelectric domain wall memory with embedded selector realized in LiNbO_3 single crystals integrated on Si wafers. *Nat. Mater.* **19**, 1188-1194 (2020).
- [54] Balke, N, *et al.* Deterministic control of ferroelastic switching in multiferroic materials. *Nat. Nanotech.* **4**, 868-875 (2009).
- [55] Matzen, S, *et al.* Super switching and control of in-plane ferroelectric nanodomains in strained thin films. *Nat. Commun.* **5**, 4415 (2014).

Q1.3: 3) thermal stability (retention) is a concern for charged domain walls. Can the authors provide some retention data, which is crucial for memory technology, to support your statement?

A1.3: We thank the Referee for raising this important issue about thermal retention, which is indeed one of the main concerns for DW memories. Following this suggestion, we have measured the current in the same place demonstrated in the initial Fig. 4f-i ('writing data' in Supplementary Fig. 13) 8 months after the DW was originally created (Supplementary Fig. 13). It is shown that the domain pattern as well as the DW preserves in the interested region. And the current signal of the

created DW still exists even though it is acceptably weaker after 8 months' storage in air. This directly indicates the strong thermal retention of the artificially created conductive DW. We also conducted erasing process to confirm that this eight-month-later conductive DW is still artificially controllable.

We have revised our manuscript and Supplementary Information to include this widely interested information.

In the main text, Page 12 (text in blue): ‘Moreover, the retention of the written conductive DW is also confirmed by a second ‘reading’ on the same sample after stored in air for eight months (Supplementary Fig. 13).’

In the Supplementary Information, c-AFM, LPFM data and the regarding erasing results after 8 months are all added in Supplementary Fig. 13.

Figure S13 | Retention and erasure of the conductive DWs on 20 u.c. BTO/Si. **a**, The written conductive DWs on 20 u.c. BTO/Si. **b**, Scanning of the created conductive DWs 8 months after creating it. Reading voltage is at +3 V. **c**, Erasing the created conductive DW by scanning over the whole region with the same negative voltage. From top to bottom: height image, LPFM phase image (arrows for polarization directions), c-AFM current image and phase-current overlapping image.

Q1.4: 4) CAFM is a two-terminal measurement that is highly contact-dependent especially for the resistive materials (in this case less than 0.5 nA at 2 volts). The behaviors observed here could be convoluted by the contact resistance.

A1.4: We thank the Referee for bringing up the important role of contact resistance, which may indeed occur in our c-AFM measurement. Following Referee's advice, we manage to lower the reading voltage with readable response from previous +4~6 V (Fig. 4, Supplementary Fig. 12) to +3~4 V (Supplementary Fig. 13b, Fig. 5) during c-AFM measurements by increasing the tip-sample contact force. In addition, the same strategy can be applied to domain switching process ('writing' and 'erasing'). As we mentioned in *A1.2* (Fig. 5), the domain switching voltage can also be lowered by optimizing the tip-surface contact (with a higher deflection setpoint).

We have revised our manuscript based on the above discussion.

In the main text, Page 10 (text in blue): 'The current signals observed here less than 0.5 nA at 2 V could be convoluted by the contact resistance, which can be improved if the measurement condition is optimized.'

Q2: In figure 2a, the out-of-plane amplitude does not look correct. For a reversed out-of-plane box-in-box switching, the amplitude channel should only have contrast at the domain walls.

A2: We thank the Referee for pointing out this confusing switching result. Ideally, we expect to obtain amplitude image which only shows contrast at DWs. The lack of contrast at DWs in this case is most likely due to the absence of the bottom electrode in the BTO/SAO/STO sample shown in Fig. 2a. Similar results have been shown in other reports [Ref 36, 50, 51].

In order to show best DWs contrast in amplitude image, we tried domain writing in many regions on the same sample and some scans yield better amplitude contrasts at DWs as shown below (Fig. A3). Even so, we note that the polarization switching is still not perfect due to the absence of bottom electrode.

We have revised our manuscript to avoid the potential confusion.

We have replaced the initial Fig. 2a with the results shown in Fig. A3 below and have also added further explanation in the main text, Page 5 (text in blue): 'Though the absence of bottom electrode and the possible static charges may lead to imperfect amplitude contrast after switching [Ref 34, 48, 49], adequate information has still been delivered.'

Figure A3 | VPFM domain switching result of 30 u.c. BTO/SAO/STO without a bottom electrode. a, Height image. **b,** Amplitude image of the switched domain. **c,** Phase image of the switched domain.

[36] Dubourdieu, C, *et al.* Switching of ferroelectric polarization in epitaxial BaTiO₃ films on silicon without a conducting bottom electrode. *Nat. Nanotech.* **8**, 748-754 (2013).

[50] Zhou, WX, *et al.* Artificial two-dimensional polar metal by charge transfer to a ferroelectric insulator. *Commun. Phys.* **2**, 125 (2019).

[51] Tian, BB, *et al.* Tunnel electroresistance through organic ferroelectrics. *Nat. Commun.* **7**, 11502 (2016).

Q3: *In figure 2, why the switching voltages are so different before and after transferring?*

A3: In Fig. 2, two hysteresis loops respectively demonstrate the out-of-plane switching voltage for as-grown BTO/SAO/STO sample and in-plane switching voltage for freestanding BTO/Si. Since the switching of polarizations relies on the effective electric field applied on the film, the presence or absence of the bottom electrode can make a huge difference of the switching voltage. The BTO/SAO/STO sample has no bottom electrode, which requires larger bias voltage to switch the polarization of the BTO film. On the contrary, the BTO/Si sample has doped silicon to be the bottom electrode, which substantially lowers the bias voltage.

To avoid this possible confusion, we have revised our manuscript for relevant clarification.

In the main text, Page 7, (text in blue): ‘**And the switching voltage difference before and after transferring is mainly due to the absence of bottom electrode in BTO/SAO/STO and the presence of it in BTO/Si.**’

Q4: *After transferring, from the c/a ratio the BTO seems to become a cubic phase (symmetry). Can the authors explain why ferroelectricity is still preserved? High-resolution RSMs shall be able to tell the in-plane tetragonal domains.*

A4: We thank the Referee for pointing out the obscurity caused by c/a ratios. In order to illustrate this point, we have performed the reciprocal space mappings (RSM) around (002) and (103) diffraction peaks of 160 u.c. BTO/Si (Fig. 3b, c), which has a similar relaxed structure as other films

but with stronger signals for analyzing. These RSM results (shown in Fig. A4 below) exhibit a mixture of a , b and c domains in a BTO/Si heterostructure. Specifically, in Fig. A4a, while the central peak at $L=2$ represents in-plane phases (long-axis lying in the plane), the component distributed at $L<2$ denotes c -axis tetragonal phase. The specific position of the peak center can be settled by multi-peak fitting. In Fig. A4b, the b -axis phase is reflected by the peak at $L=3$ and $H>1$, with the a -axis phase corresponding to the peak at $H<1$ and c -axis phase corresponding to the peak at $L<3$. Besides proving the presence of three tetragonal phases, these two RSM images further demonstrate much stronger peak intensities for in-plane phases, suggesting the dominance of a -axis and b -axis tetragonal phases in the structure of freestanding BTO membranes. Meanwhile, the PFM results and complementary HADDF-STEM images of BTO/Si (Supplementary Fig. 9) also supports the dominance of a -axis and b -axis tetragonal phases.

With the above information, we realized that the lattice constant a and the c/a ratios offered in initial Fig. 3 are actually misleading. As the diffraction intensity of thinner films are too weak to run RSM scans around (103) reflection, lattice constants c and a are extracted from the 2θ - ω scans along (002) and (103) reflections. The actual obtained c and a mainly reflect the c and mixed a values for the dominated in-plane tetragonal phases, including both a -axis and b -axis tetragonal phases. As c -axis is the short axis in both two phases, the obtained c values indeed reflect the dominant c of the sample. However, as a -axis is the short axis in b -axis tetragonal phase but the long axis in a -axis tetragonal phase, the obtained a values are actually collective results of indistinguishable a -axis domain and b -axis domain. Therefore, the c/a ratios being close to 1.00 is actually not a sign of cubic phase, but a collective result from different tetragonal phases. The RSM images, together with other evidence, indicates that the in-plane phases dominate the structural phase composition of the sample.

As the Referee suggested, we have revised our manuscript and Supplementary Information to provide further explanation.

In the main text, we have replaced the original Fig. 3b with RSM images and have also deleted the misleading a values for freestanding BTO in Fig. 3a, as they are actually collective results from a -axis and b -axis domains.

In the main text, Page 8 (text in blue): ‘In Fig. 3a, freestanding BTO membranes of different thicknesses all present sharp reductions in out-of-plane lattice constant c values compared with as-grown films, indicating general crystal structure collapses in c direction. In Fig. 3b and 3c, the RSM images around (002) and (103) diffraction peaks for BTO/Si suggest the structural composition of three different tetragonal phases (denoted by arrows in Fig. 3b, c).’

In the main text, Page 8 (text in blue): ‘Therefore, through the transferring process, under the cooperative effects of strain relaxation and depolarization field, BTO membranes transform from c -axis tetragonal phase to a mixture of a -axis, b -axis and c -axis tetragonal phases to stabilize ferroelectric polarization, as schematically illustrated in Supplementary Fig. 8c. It is also noteworthy that the peak intensities of a -axis and b -axis tetragonal phases in RSM images (Fig. 3b, c) are much

stronger than that of *c*-axis phase, manifesting the dominance of in-plane tetragonal phases among the whole phase structures. Since each tetragonal phase (*a*-axis, *b*-axis or *c*-axis) possesses polarization orientation along their elongated axis, the dominant formation of *a*-axis and *b*-axis tetragonal phases thus accounts for the emergence of in-plane ferroelectricity throughout the freestanding BTO membrane, which is also consistent with our phase-field simulation result.’

In the Supplementary Information, schematics of structural composition of BTO membrane on Si (shown below as Fig. A5) has been added as Supplementary Fig. 8c.

In the Supplementary Information, HADDF-STEM images of 160 u.c. BTO/Si are shown in Supplementary Fig. 9.

Figure A4 | Structural analysis of a freestanding 160 u.c. BTO membrane on Si. a,b, RSM around (a) (002)- and (b) (103)-diffraction peaks. The mappings are based on a cubic structure with lattice constant of 0.4 nm. r.l.u., relative light units. Different parts of the peaks represent different domains, illustrated in detail by the arrows.

Figure S9 | In-plane ferroelectricity of BTO membrane. **a,b,** Maps of polar atomic displacement vectors obtained from the plan-view HAADF-STEM images of a 160 u.c. BTO membrane transferred to a holey carbon TEM grid. **c,d,** Zoom-in atomically resolved HAADF-STEM images. The average displacement of Ti atom is about 21 pm for ferroelectric domain in **a** and 17 pm for domain in **b**.

Figure A5 | Schematics of structural composition of BTO membrane on Si, illustrating the mixed a-axis, b-axis and c-axis tetragonal phases.

Q5: The authors mentioned the lack of screening results in a large depolarization field. I am wondering did not the author use Au/Si wafer as the bottom electrode for CAFM measurements? There should be enough free electrons for screening from the Au. Can the author comment on it?

A5: In the BTO/SAO/STO heterostructure, the out-of-plane polarization is maintained due to the compressive epitaxial strain imposed by the substrate, though the depolarizing field tends to suppress it. However, when SAO layer starts to dissolve, the compressive strain disappears instantly but the depolarizing field still exists, which destabilizes the pristine out-of-plane polarization. Consequently, the in-plane tetragonal phases form immediately when the transferring (SAO dissolution) process starts. In other words, the in-plane multidomain structure has already formed before the BTO membrane is well attached to the Au-coated Si. Our phase-field simulation also confirmed that the in-plane multidomain structure emerges in freestanding BTO when the epitaxial strain is released.

To address this question, we have revised the manuscript to make further explanation on this point.

In the main text, Page 8 (text in blue): ‘Though compressive strain prevails over depolarizing field in BTO/SAO/STO, when the transferring process starts, depolarizing field also becomes influential in freestanding BTO as strain starts to be released.’

In the main text, Page 9 (text in blue): ‘Note that the in-plane multidomain structure of BTO/Si is formed the moment transferring process starts (before freestanding BTO touches the Au-coated Si), therefore the possible screening effect from Au doesn’t affect the formation of these in-plane ferroelectric domains.’

Q6: In figure 4b, the blue line is denoted as "off DW" showing a constant current level at even zero bias. The current level on DW is even lower at zero bias. Can the author comment on it?

A6: We are grateful that the Referee points out the confusing ‘Off DW’ data here. This actually comes from an extrinsic effect of capacitor-like charging and discharging behavior during the rapid I-V loops. For clarification, we demonstrate an intact ‘Off DW’ I-V hysteresis (Supplementary Fig. 17), which suggests that the behavior of I-V curves measured at insulating domain area (‘Off DW’) just like a capacitor. This charging and discharging loop is caused by the rapid switching speed of applied AC voltage. As a result, a current offset of $\sim \pm 10$ pA at zero bias was observed. And this is why the initially presented ‘Off DW’ I-V curve is having a constant current at zero bias while the current level ‘On DW’ is even lower at zero bias.

Despite the confusing current level at zero bias, the distinct contrast between ‘On DW’ and ‘Off DW’ currents is not affected, as the ‘Off DW’ current maximum is repeatably obtained at the level of 10 pA. To weaken the influence of charging and discharging behavior, I-V curve of ‘Off DW’ are retested with a lower sweep frequency at only positive bias region.

Based on the Referee’s comment, we have revised our manuscript as well as Supplementary Information to specify this point.

In the main text, we have replaced our original ‘Off DW’ curve in Fig. 4b with optimized ‘Off DW’ curve to avoid the confusion.

In the main text, Page 16 (text in blue): ‘In order to weaken the influence of charging and discharging behavior when measuring I-V curves (Supplementary Fig. 17), I-V data can be obtained with a lower sweep frequency at only positive bias region.’

In the Supplementary Information, Off DW hysteresis is shown in Supplementary Fig. 17.

Figure S17 | I-V curves off and on DW. **a**, I-V curves on and off the conductive DWs updated in Fig. 4b. **b**, ‘Off DW’ I-V hysteresis.

Q7: In figure 4c and 4i, not all the H-t-H walls are showing conductive behaviors. For example, at the bottom right of figure 4c, there should be also an H-t-H domain wall according to the author's remarks, but it seems not conductive or at least not as conductive as the marked ones. Can the author comment on it?

A7: We thank the Referee for pointing out this special case in Fig. 4c. Though we generally expect all the H-H DWs to be conductive, the actual conductivity of the DW strongly relies on the strictly H-H component of polarizations at two sides of the DW. For the inclined DWs that are not so strictly perpendicular to the polarization, the lower band bending results in a lower concentration of charge carriers compared to those at the perpendicular DWs. For the extreme case, when the domain wall is parallel to the polarization, there is basically no band bending and negligible charge accumulation at the DWs. At the bottom right of Fig. 4c, the DW is inclined to the polarization (about 40~45° here), which explains the absence of conductive behaviors.

To address this question, we have revised our manuscript for clarification.

In the main text, Page 11 (text in blue): ‘Additionally, since the band bending and charge accumulation strongly depend on alignment of the DW with the polarization direction, only those H-H DWs nearly perpendicular to the polarization direction present prominent conductivity (Fig. 4e).’

Q8: In figure S1, it does not look like 6 u.c. deposition of SAO to me.

A8: SAO has a cubic structure with the lattice constant a of 15.844 Å, which is almost four times of STO lattice constant a of 3.905 Å ($a_{SAO} \approx 4a_{STO}$), as shown in Fig. A6. It is reported that for RHEED oscillations of SAO [Ref 37, 49], the atomic sublayers generate four periods of intensity peak in the growth of one SAO unit cell. From the oscillation pattern in Supplementary Fig. 1, the yellow vertical lines denote the deposition of 6 unit-cell SAO layers. Note that the first two SAO unit cells contain fewer periods of intensity peaks, since the beam flux is slightly unstable at the beginning of

SAO deposition.

Based on the Referee's comment, we have revised the Supplementary Information.

In the Supplementary Information, Supplementary Fig. 1 has been updated by marking the unit cell growth of SAO layer.

Figure S1 | Freestanding BaTiO_3 (BTO) films preparation. **a**, RHEED oscillation curves for 20 u.c. BTO thin film grown on STO substrate with 6 u.c. SAO sacrificial layer. Insets are corresponding RHEED diffraction patterns. Yellow vertical line indicates the growth of one unit-cell SAO layer. **b**, AFM image for as-grown 20 u.c. BTO/SAO/STO heterostructure showing clear steps and terraces. **c**, Schematics of clamping device used for fixing the heterostructure and Si substrate during transferring process.

Figure A6 | Schematic of the crystal structure of $\text{Sr}_3\text{Al}_2\text{O}_6$ from the side view and top view. Oxygen atoms are omitted for simplicity [Ref 49].

[37] Lu, D, *et al.* Synthesis of freestanding single-crystal perovskite films and heterostructures by etching of sacrificial water-soluble layers. *Nat. Mater.* **15**, 1255-1260 (2016).

[49] Sun, HY, *et al.* Epitaxial optimization of atomically smooth $\text{Sr}_3\text{Al}_2\text{O}_6$ for freestanding perovskite films by molecular beam epitaxy. *Thin Solid Films* **697**, (2020).

In conclusion, this paper reported the head-to-head conducting domain walls of BaTiO₃ films on Si, which is not surprising for the ferroelectrics. Some technical details can be improved or clarified. But I found it is hard to convince me this article has significant breakthrough from the previous studies (Science advances 3, no. 6 (2017): e1700512., Nature nanotechnology 13, no. 10 (2018): 947-952., Nature nanotechnology 10, no. 2 (2015): 145-150., and Advanced Materials, 32(9), p.1905132.).

We agree with the Referee that conductive DW memories as the promising candidate for high-density and low power nonvolatile memories have been a hot interest and intensively investigated since the R. Ramesh's group reported the conductive DW in BFO in 2009. However, previous works usually demonstrate perovskite-based DW memories with as-grown in-plane ferroelectricity in thin films clamped by their perovskite oxide substrates. As discussed in *AIJ*, epitaxial growth of perovskite oxides directly on Si substrates has long been hindered by issues like interface oxidation and large lattice mismatch, only a very limited number of perovskite oxides such as SrTiO₃ have been demonstrated to be able to synthesized on Si using a very sophisticated and time-consuming method [Ref 34]. Here, our work makes a breakthrough in integrating high-quality crystalline ferroelectric BaTiO₃ membranes on Si using a growth and transfer method. An intriguing polarization switching takes place in freestanding BTO films due to the collective effects of strain-relaxation and depolarization field, which agrees with our phase-field simulation. We further demonstrate the realization of rewritable conductive DWs in this heterostructure, providing preliminary basis for future device applications. As such, our work provides the first example of the integrating conductive DWs in ferroelectric perovskite oxides onto Si, inspiring the community to make more systematic studies in integrating the rich functionalities of perovskite oxides with Si for real electronic applications in the near future.

[34] Warusawithana, MP, *et al.* A Ferroelectric Oxide Made Directly on Silicon. *Science* **324**, 367 (2009).

Responses to Reviewer #3:

The authors report on the fabrication and switching of freestanding ferroelectric BaTiO₃ (BTO) membrane integrated on a Si wafer. By releasing the epitaxial tetragonal BTO film using the Sr₃Al₂O₆ buffer layer, the initially c-axis polarized film transforms into a mixture of a-, b-, and c-axis polarized domains, exhibiting an apparent cubic structure. The released BTO membrane therefore show in-plane polarization which can be switched via vertically applied bias functioning as a memory element.

Freestanding single crystalline oxide membranes are one of the growing areas of research not only from academic but also from industrial perspectives. Ferroelectrics, which is one of the major properties of complex oxides, is especially an attractive target for membrane applications.

Overall, this paper is well organized but I have a few questions regarding the experiments and interpretations, which I would like the authors to address before considering further.

We thank the Referee #3 for the positive comments and constructive suggestions. Below we address Referee's questions point by point and revise our manuscript accordingly.

Q1: In Figs. 4f and 4g, the authors show domain switching in a very small area. Do the authors have larger area maps before/after switching? Given the emphasis on application, I think the actual scale of the membrane as well as the polarization map would be very instructive.

AI: We are grateful for the Referee's suggestion and we fully agree that the scale of domain switching is of great concern. Following the Referee's suggestion, we have performed more PFM scans on a largest area as shown below (Supplementary Fig. 11a). As the whole BTO/Si sample exhibits in-plane ferroelectricity, we conducted domain switching results in two randomly picked regions of a 20 u.c. BTO/Si sample (Supplementary Fig. 11b, c), where larger size H-H DWs are both created through 'writing' process. In addition, Fig. 5 also displays the created larger size H-H DW in 15 u.c. BTO/Si. All these results indicate that large size H-H DW can be achieved in any regions throughout the whole sample, addressing the concern of the scale (both the amount and the size) of DWs. As for the actual size of the membranes, we can transfer complete sample up to 5mm×5mm or larger, as also reported in previous literatures [Ref 40, 61].

We have revised our manuscript and Supplementary Information to demonstrate the above supportive results.

In the main text, Page 11 (text in blue): 'By scanning the tip over two regions with opposite biases (Supplementary Fig. 11d), we were able to create opposite polarization orientations in neighboring domain by the electric trailing field of the tip [Ref 54, 55], and thus artificially create large-size H-H conductive DWs randomly over the membrane (Supplementary Fig. 11).'

In the main text, domain-writing results on 15 u.c. BTO/Si are displayed in Fig. 5. Corresponding

illustration is also added in the main text, Page 11 (text in blue): ‘Fig. 5b and Fig. 5e respectively demonstrate the domain structures before and after domain switching (‘writing’) on 15 u.c. BTO/Si. Evident H-H DWs appeared after switching, and was verified to be conductive in the later c-AFM measurement (Fig. 5f).’

In the Supplementary Information, creation of large size H-H DW in random regions are demonstrated in Supplementary Fig. 11.

Figure S11 | Scale of H-H DWs. **a**, LPMF results from large to small area, showing that in-plane ferroelectricity is exhibited throughout the whole sample. **b,c**, Large size H-H DW created on two random regions, showing that large size H-H DW can be created on BTO/Si in both large amount and large size. **d**, Scanning method for creating H-H DWs using trailing field.

Figure 5 | Writing and erasing conductive DWs in 15 u.c. BTO/Si. **a-c**, Pristine state of the tested region. From top to bottom: AFM height image, LPFM phase image, c-AFM current image (same for **d-l**). **d-f**, Results after ‘writing’ process. **g-i**, Results after ‘erasing’ process which switches the H-H DW into T-T DW, scanning method shown in **m**. **j-l**, Results after ‘erasing’ process which switches the whole region into monodomain, scanning method shown in **n**. **o**, I-V curves measured at different positions.

[40] Ji, D, *et al.* Freestanding crystalline oxide perovskites down to the monolayer limit. *Nature* **570**, 87-90 (2019).

[54] Balke, N, *et al.* Deterministic control of ferroelastic switching in multiferroic materials. *Nat. Nanotech.* **4**, 868-875 (2009).

[55] Matzen, S, *et al.* Super switching and control of in-plane ferroelectric nanodomains in strained thin films. *Nat. Commun.* **5**, 4415 (2014).

[61] Zhang, B, Yun, C, MacManus-Driscoll, JL. High Yield Transfer of Clean Large-Area Epitaxial Oxide Thin Films. *Nanomicro Lett.* **13**, 39 (2021).

Q2: Also, I am concerned about the concentrated electric field lines close to the c-AFM tip when 6V is applied on the entire back side of the membrane. The area ratio must be gigantic given the small radius of curvature of the AFM tip. Consequently, wouldn't a large electric be applied locally around the tip, which readily switches the domains and leave parts of the membrane far from the tip unswitched?

A2: Thanks to Referee's valuable question. As the Referee pointed out, it is truly possible that such reading voltages lead to unwanted domain switching. To confirm that such reading voltages are not destructive to the domain structure, we scanned the area of DWs before and after c-AFM measurements in 20 u.c. BTO (after 8 months, Supplementary Fig. 13) and 15 u.c. BTO (Fig. A2). From two sets of LPFM phase images taken on different samples, one can discern that DWs have not been affected by the c-AFM scanning in our experiments. Therefore, though the hysteresis loop demonstrated in Fig. 2 shows coercive field (voltage) lower than 4 V, the actual voltage to switch polarization probably needs to be larger than the detected coercive field (voltage). Such voltage differences may be resulted from the difference in tip conditions during measurements, as we use different tips for LPFM measurements and c-AFM measurements. Supplementary Fig. 6 demonstrates the domain switching results by using two different tips, with switching voltage being much lower when using new tip than old tip, probably because old tip is more worn (tatty) than new tip and thus has a worse contact. In short, we have confirmed that the domains are not switched during our 'reading' process, which may be due to the different tip conditions for 'writing' and 'reading'.

To avoid the possible domain switching as the Referee suggested, we thus choose to use new tips for both 'writing' and 'reading' process on 15 u.c. BTO/Si (Fig. 5), with reading voltage being 3.5 V, much lower than 6 V writing voltage. Such voltage difference guarantees that the c-AFM 'reading' process won't do harm to the existed domain, as evidenced by our LPFM measurements before and after c-AFM scanning (Fig. A2).

We have revised our manuscript and Supplementary Information following Referee's advice.

In the main text, Page 7 (text in blue): 'Notice that the actual switching voltage during scanning is dependent on the tip-surface contact, which is determined by both the deflection setpoints (Supplementary Fig. 5) and the tip conditions (Supplementary Fig. 6).'

In the main text, Page 16 (text in blue): 'LPFM phase images before and after c-AFM measurements have been tested to ensure conductive tips being nondestructive to existed domain structures.'

In the Supplementary Information, we have demonstrated the AFM and LPFM phase images of 20 u.c. BTO/Si in the original version (before c-AFM measurement) and 8 months after c-AFM measurement in Supplementary Fig. 13 (first two columns).

In the Supplementary Information, domain switching results by using different tips are shown in Supplementary Fig. 6.

Figure S13 | Retention and erasure of the conductive DWs on 20 u.c. BTO/Si. a, The written conductive DWs on 20 u.c. BTO/Si. **b,** Scanning of the created conductive DWs 8 months after creating it. Reading voltage is at +3 V. **c,** Erasing the created conductive DW by scanning over the whole region with the same negative voltage. From top to bottom: height image, LPFM phase image (arrows for polarization directions), c-AFM current image and phase-current overlapping image.

Figure A2 | Domain patterns before and after c-AFM measurement on 15 u.c. BTO/Si. a, Phase image before c-AFM scanning. **b,** c-AFM current image. **c,** Phase image after c-AFM scanning.

Figure S6 | Domain switching results by using different tips. a, Domain can be switched when switching voltage reaches +6 V by using new tip. **b**, Trailing field method of domain switching. **c**, Domain can be completely switched until switching voltage reaches +9 V by using old tatty tip.

Q3: How were the direction of the polarization determined in Fig. 4d, 4f, and 4g, i.e., the red and blue arrows? Also, in Figs. 4e, 4i, the overlapping of the current map to the LPMF map are not convincing. The authors should measure areas with markers active for both maps to identify that the measurements were taken on the same area.

A3: The red and blue arrows are determined by domain writing process (trailing field method, [Ref 54, 55]), as show in Supplementary Fig. 10. For the region that shows intrinsic conductive DWs (Fig. 4c-e), in order to keep the domain pattern undestroyed in the target region for future examination, we first do LPMF scan for a much larger area that contains the target region. Then we conduct domain writing process for a region (test region) in the large area but far from the target region. In the end, we re-examine the LPMF of the large area. Since we have knowledge of polarizations in the test (written) region, we can easily recognize the polarization directions that the two phase-image-colors represent in the whole large area, thus we can also determine the polarizations in the target region. For the region shown in initial Fig. 4f-i, since conductive DWs is artificially created (instead of intrinsic) in this region, we can determine the polarization directly after we complete domain switching on it.

In order to ensure that we conduct c-AFM and LPMF scan at the same target region, we locate the target region by scanning a $30 \times 30 \text{ um}^2$ (later followed by $10 \times 10 \text{ um}^2$ and $6 \times 6 \text{ um}^2$ etc.) area that has the same center with it, as scanning probe microscopy (SPM) is easy to rescale the scanning size without shifting the scanning center. When we shift measurement mode, we find the large region first, since large regions usually have topography features distinct enough. Recalling that our target region has the same center with the large region, we then gradually scale down the scanning size to eventually find the target region. As tiny drifting may occur during the scanning, we do need special

markers in AFM images to help adjust the center positions, so that we can accurately find our target regions.

Moreover, as discussed in *AI*, we also performed similar writing and reading experiments in films with different thickness, such as 15 u.c. sample shown in Fig. 5. The AFM images and the overlapped DW positions in Supplementary Fig. 12 and Fig. 5 all show that our ‘writing’ and ‘reading’ are conducted at the same region. The AFM images for the region shown in Fig. 4c-e are also demonstrated in Fig. A7 as evidence.

Following Referee’s advice, we have revised our manuscript and Supplementary Information for further clarification.

In the main text, Fig. 5 has included the AFM images corresponding to ‘writing’, ‘reading’ and ‘erasing’ process (demonstrated above in *AI*) to confirm that the measurements were taken at the same area.

In the main text, Page 10 (text in blue): ‘AFM images are confirmed to guarantee that measurements are taken on the same region. Polarization orientations for in-plane domains were verified by domain switching based on trailing field method [Ref 54, 55] (Supplementary Fig. 10).’

In the Supplementary Information, Supplementary Fig. 10 is added to demonstrate the process of determining polarization direction for regions shown in Fig. 4c-e.

In the Supplementary Information, AFM images corresponding to the current image and phase images have been demonstrated in Supplementary Fig. 12 for 20 u.c. BTO DW writing process.

Figure S10 | Polarization direction determination of the region shown in Fig. 4c-e. **a**, The test region before (left) and after (right) domain writing. The center red box is written to be red-arrow polarized by trailing field. **b**, Schematic of using trailing field method for domain switching. **c**, Based on the information in the green box (the whole test region is marked as green in the right panel of **a**), black phase-color represents red-arrow polarization and yellow phase-color represents

blue-arrow polarization. The blue box ensures the consistency of left panel area and right panel area in c. The target region is marked as brown in right panel of c. **d**, Polarization result of the target region shown in Fig. 4c-e based on results in c.

Figure S12 | ‘Writing’ and ‘reading’ H-H DW on 20 u.c. BTO/Si. **a,b**, AFM height image (a) and LPFM phase image (b) before ‘writing’. **c,d**, AFM height image (c) and LPFM phase image (d) after ‘writing’. **e,f**, AFM height image (e) and c-AFM current image (f) by ‘reading’ process.

Figure A7 | AFM images for regions shown in Fig. 4. **a,b**, AFM images during LPFM (a) and c-AFM (b) of the region shown in Fig. 4c-e.

[54] Balke, N, *et al.* Deterministic control of ferroelastic switching in multiferroic materials. *Nat. Nanotech.* **4**, 868-875 (2009).

[55] Matzen, S, *et al.* Super switching and control of in-plane ferroelectric nanodomains in strained thin films. *Nat. Commun.* **5**, 4415 (2014).

Q4: In Fig. 4b, the current observed for on-DW shows a few nA. However, in the current maps of Fig. 4e, 4i, the scale bar shows up to 160 pA, an order of magnitude smaller than expected at 4V. I would like the authors to first show statistics and secondly to explain if there is possibility of amorphous or non-ferroelectric insulating regions acting as additional resistance. BTO is known to be hydroscopic and even though their film may be single crystalline, 12h immersion in water may selectively dissolve the surface Ba ions, leaving TiO₂ or Ba-deficient BTO in the membrane.

A4: We thank the Referee for the comment, which makes us realize that the original small range of the color bar may cause a misleading impression for readers. The original scale bar is provided for a distinction of the insulated region and conductive DWs, based on the data of the whole region. The actual current at the conductive DWs is not limited by 160 pA. We have adjusted the scale bar to offer a better sense of current amplitude (Fig. A8a). Two current line profiles of the conductive DWs (Fig. A8b) reveal that current can both reach ~12 nA on DW while the insulating region is at a level of 10 pA.

In addition, we are thankful that Referee points out the potential influence of water immersion. We agree that our transferring process may cause Ba-deficiency in BTO and thus may potentially bring amorphous regions, which act as additional resistance during the measurements.

We have revised our manuscript following Referee’s suggestions.

In the main text, color bar of Fig. 4c is adjusted and current line profiles across the conductive DW (shown below in Fig. A8b) are added as insets in Fig. 4c.

In the main text, further illustration is provided in Page 10 (text in blue): ‘Phase pattern in Fig. 4d displays distinct DWs, the regarding conductivities of which are demonstrated in c-AFM images (Fig. 4c) with representative current line profiles shown in insets.’

Figure A8 | Current signals of conductive DWs. a, Fig. 4c c-AFM current image. **b,** Current amplitude along the line sections drawn in **a**.

Q5: By switching the polarization, can the authors show that the initially H-H DW turns into T-T DW by taking I-V cures before/after switching?

A5: Following Referee’s suggestion, we erased H-H DW in 15 u.c. BTO/Si by switching it into T-T DW (Fig. 5). As we presented in main text, only H-H DWs are conductive in freestanding BTO/Si heterostructure [Ref 8, 13, 53, 56]. In our erasing process, consistent results are demonstrated that H-H DW shows distinct current signals while T-T DW is insulated.

We have also revised our manuscript and Supplementary Information to include this erasing experiment.

In the main text, we have demonstrated the main erasing results in Fig. 5 (demonstrated above in **A1**) and added corresponding explanation in Page 11 (text in blue): ‘When it comes to erasing the

conductive DW, we can either turn it into an insulated T-T DW (Fig. 5h, m) or directly switch the whole region into a monodomain (Fig. 5k, n) to eliminate the DW. Both methods have proved to be effective in ‘turning off’ the conductivity (Fig. 5i, l, o).’

[8] Sluka, T, Tagantsev, AK, Bednyakov, P, Setter, N. Free-electron gas at charged domain walls in insulating BaTiO₃. *Nat. Commun.* **4**, 1808 (2013).

[13] Meier, D, *et al.* Anisotropic conductance at improper ferroelectric domain walls. *Nat. Mater.* **11**, 284-288 (2012).

[53] Sharma, P, *et al.* Conformational Domain Wall Switch. *Adv. Funct. Mater.* **29**, 1807523 (2019).

[56] Farokhipoor, S, Noheda, B. Conduction through 71 degrees domain walls in BiFeO₃ thin films. *Phys. Rev. Lett.* **107**, 127601 (2011).

Q6: The authors show a schematic of the potential landscape in Fig. 3c-f. The x-axis, which I suspect is the polarization along [001] or [010], does not have an origin. Therefore, the asymmetry in Fig. 3c is not obvious. I assume the dotted vertical line is $P = 0$, but the authors should clarify.

A6: We thank the Referee for pointing out this vagueness in our text.

We have revised our manuscript by adding the origins into Fig. 3c-f, as shown below.

Figure 3 | Structural transition of BTO. a, Lattice constants of BTO at different thicknesses before and after transferring. Bulk BTO lattice constants are offered for comparison in **a**. **b,c,** RSM around **(b)** (002)- and **(c)** (103)-diffraction peaks. The mappings are based on STO reciprocal lattice unit

(r.l.u.). Different parts of the peaks represent different domains, illustrated in detail by the arrows. Error bars are drawn considering instrument precision and accidental errors. **d,e**, Qualitative potential diagram along [001] for BTO/SAO/STO in **d** and BTO/Si in **e**. Out-of-plane polarization directions are shown for both cases to clarify the *c*-axis polarization variation. **f,g**, Qualitative in-plane potential diagrams for BTO/SAO/STO in **f** and BTO/Si in **g**.

Q7: LL 114-117. Do the authors have evidence to support this claim? Is Fig. 2a vs. 2c? I would like the authors to be more quantitative in their description.

A7: We are grateful to the Referee for pointing out this unsubstantiated claim. We agree that our data is not supportive enough to make such statement since the LPFM amplitude result is determined by multiple factors including polarization and thickness of the sample. Instead, from our RSM images of BTO/Si (Fig. 3b, c), the intensities of peaks around *a*-axis and *b*-axis tetragonal phase are distinctively larger than those of *c*-axis phase, suggesting the dominance of in-plane ferroelectric phases in the overall phase composition. The HADDF-STEM images (Supplementary Fig. 9) also demonstrate prominent *a*-axis and *b*-axis ferroelectric domains in freestanding BTO membranes. In short, we confirm the predominance of in-plane domains, but may not evidence the dominance of in-plane polarization amplitude.

We have revised the manuscript as well as Supplementary Information based on the above discussion.

In the main text, we have provided RSM images of BTO/Si in Fig. 3 (demonstrated above in **A6**) and have revised our statement in Page 8 (text in blue)

Original version: ‘It is also noteworthy that the amplitudes of LPFM scanning results are even larger than those of VPFM scanning at the same driving voltage, evidencing the strong in-plane polarization in ferroelectric phase of BTO/Si.’

Revised version: ‘It is also noteworthy that the peak intensities of *a*-axis and *b*-axis tetragonal phases in RSM images (Fig. 3b, c) are much stronger than that of *c*-axis phase, manifesting the dominance of in-plane tetragonal phases among the whole phase structures. Since each tetragonal phase (*a*-axis, *b*-axis or *c*-axis) possesses polarization orientation along their elongated axis, the dominant formation of *a*-axis and *b*-axis tetragonal phases thus accounts for the emergence of in-plane ferroelectricity throughout the freestanding BTO membrane, which is also consistent with our phase-field simulation result.’

In the Supplementary Information, HADDF-STEM images in Supplementary Fig. 9 shows prominent in-plane polarizations.

Figure S9 | In-plane ferroelectricity of BTO membrane. **a,b,** Maps of polar atomic displacement vectors obtained from the plan-view HAADF-STEM images of a 160 u.c. BTO membrane transferred to a holey carbon TEM grid. **c,d,** Zoom-in atomically resolved HAADF-STEM images. The average displacement of Ti atom is about 21 pm for ferroelectric domain in **a** and 17 pm for domain in **b**.

REVIEWER COMMENTS

Reviewer #1 (Remarks to the Author):

The authors addressed all points I mentioned.

Reviewer #2 (Remarks to the Author):

I would like to thank the authors for their efforts to improve the quality of their data and for considering my comments. Below, I respond to individual answers of the authors only where necessary. The numbering refers to the question from the first review.

Q1.1. Thank the authors for the detailed explanation and the future device study proposal. Technically, I have no doubt the proposed in-plane device would work, but to have this device be commercially competitive, the four-terminal device will be way too big for the footprint. Also, given the distance between the electrodes, which is limited by photolithography, one has to apply extremely high voltage. For example, from Fig. 2d, the coercive field is about $4V/30\text{nm} \sim 3.3\text{MV/cm}$, which by the way is still an incredibly high electric field, and translating this field into the in-plane electrode we can estimate the voltage needed for creating domain walls that author proposed. Assuming the spacing between electrodes can be made around 100-nm by e-beam lithography, then the voltage will be not suppliable in the integrated circuit.

Q1.3 Thank the author's effort to clear the retention concern of mine. However, I found the images presented in S Fig. 13 deepen my worries for the following reasons: 1) the domain structure does not look the same to me from panels a and b. I fully understand it is not easy to find the very same spot especially when you have features at this scale. The arrow-pointed domain walls at least in my opinion have concave and convex features indicating with time/thermal perturbation the domain wall will move. 2) in the conducting current channel, the hot spots are not appearing at the same position as well, it is clearer when the authors imposed the conducting map with the domain images showing at the bottom panel. The arrow-pointed in the "writing" column shows no measured current while the "retention" column shows $\sim 4\text{ pA}$ conducting level. Would not this imply the unpredictable nature of this type of charged domain walls?

Q7. The authors mentioned an important point of these head-to-head domain walls in this study. That is the angular dependence of domain wall conducting behavior, which echoes my concerns. This delicate domain structure is not suitable for practical application. Can the authors show the angular dependency of domain wall conductance, such as the angle between polarization versus conducting current at a certain tip bias?

Reviewer #3 (Remarks to the Author):

The authors have addressed my concerns with additional experiments, simulations, and significant changes to the manuscript which have made their claims clearer. As the other two

reviewers point out, the feasibility of this strategy toward CMOS compatible memory device remains a challenge in my opinion. However, I believe the current work does provide an interesting platform for which to make progress on and would recommend publication in Nature Communications in the present form.

We appreciate all the referees for the very positive and constructive comments and recommendations of publication. Meanwhile, Referee #2 still has a few additional questions that he/she would like us to clarify. In the following, we discuss all Referee #2's questions point by point and have revised our manuscript and Supplementary Information accordingly.

Responses to Reviewer #2:

I would like to thank the authors for their efforts to improve the quality of their data and for considering my comments. Below, I respond to individual answers of the authors only where necessary. The numbering refers to the question from the first review.

We thank Referee #2 for the positive review of the previous rebuttal. Below we address the further concerns and questions point by point.

***Q1.1:** Thank the authors for the detailed explanation and the future device study proposal. Technically, I have no doubt the proposed in-plane device would work, but to have this device be commercially competitive, the four-terminal device will be way too big for the footprint. Also, given the distance between the electrodes, which is limited by photolithography, one has to apply extremely high voltage. For example, from Fig. 2d, the coercive field is about $4\text{V}/30\text{uc} \sim 3.3\text{MV}/\text{cm}$, which by the way is still an incredibly high electric field, and translating this field into the in-plane electrode we can estimate the voltage needed for creating domain walls that author proposed. Assuming the spacing between electrodes can be made around 100-nm by e-beam lithography, then the voltage will be not suppliable in the integrated circuit.*

A1.1: We thank the Referee for considering our in-plane device would technically work while raising up the concern about the size and switching voltage.

Regarding the switching voltage, it is actually much lower than the value estimated by the Referee. Although Fig. 2d indicates a coercive field of about 4 V for a 30 u.c. film, the real coercive field of the film is not $\sim 3.3\text{ MV}/\text{cm}$ ($4\text{ V} / 30\text{ u.c.}$) since this nominal value strongly depends on how close the tip contacts the film as we have discussed in the previous version of the rebuttal. Also, given

that the voltage is applied by a conductive tip over the sample with the bottom electrode grounded, there are both vertical and lateral components of the electric field and the effective in-plane electric field for polarization switching should be smaller. In the four-terminal model, the voltage is applied by coplanar terminals, and the electric field lines are mainly in the plane of BTO membrane, meaning that the applied voltage can be much more effective for in-plane polarization switching. In fact, a recent report shows that 12.5 nm BTO thin film can have a coercive field of ~ 400 kV/cm [Ref 62], which is equivalent to 0.5 V switching voltage. In our preliminary trial with two-terminal device with ~ 200 nm gap on our BTO/Si, a bias voltage of 2 V is enough to create a new domain wall (Fig. A1). Based on above information, the switching voltage for the four-terminal model with 100 nm or narrower electrode spacing (common for the CMOS lithography technique) should be suppliable in the integrated circuit. As to the size of the four-terminal model, this 4-terminal design is much smaller than the commercialized FRAM based on 1T1C structure ($\sim 0.5 \mu\text{m}^2$) [Ref 63].

Moreover, while we agree that there are still technical challenges in order to make this new device to work and be commercially competitive, the main contribution of this work is the observation of in-plane ferroelectricity in relaxed ultrathin BTO membranes and the successful integration of conductive ferroelectric DWs on silicon, highlighting the promising integration of functional ferroelectric oxides with silicon for novel electronic devices.

Figure A1 | Creating DW with two-terminal model. a,b, Initial LPFM amplitude (a) and phase (b) images. c,d, LPFM amplitude (c) and phase (d) images after applying +2 V bias between metal

terminal#1 and terminal#2. Arrows demonstrate polarization orientations.

[62] Jiang, Y., Parsonnet, E., Qualls, A. *et al.* Enabling ultra-low-voltage switching in BaTiO₃. *Nat. Mater.* (2022).

[63] Udayakumar, KR, *et al.* Manufacturable High-Density 8 Mbit One Transistor–One Capacitor Embedded Ferroelectric Random Access Memory. *Jpn. J. Appl. Phys.* **47**, 2710 (2008).

Q1.3: *Thank the author’s effort to clear the retention concern of mine. However, I found the images presented in S Fig. 13 deepen my worries for the following reasons: 1) the domain structure does not look the same to me from panels a and b. I fully understand it is not easy to find the very same spot especially when you have features at this scale. The arrow-pointed domain walls at least in my opinion have concave and convex features indicating with time/thermal perturbation the domain wall will move. 2) in the conducting current channel, the hot spots are not appearing at the same position as well, it is clearer when the authors imposed the conducting map with the domain images showing at the bottom panel. The arrow-pointed in the “writing” column shows no measured current while the “retention” column shows ~4 pA conducting level. Would not this imply the unpredictable nature of this type of charged domain walls?*

A1.3: We thank the Referee for raising this concern. 1) The concave and convex features come out as a result of our measurement settings. For the ‘writing’ measurement, we adopted a smaller scan step size, therefore we have more pixels for the whole region. For the ‘retention’ measurement, we used a larger scan step size (or less pixels for the whole region) by default, which therefore limits the smoothness of the domain walls shown in phase image. Since we have erased the structure in taking the data shown in panel c, we could not run high resolution scan to replace the data in panel b. Moreover, as the very detailed features in PFM data have certain dependences on tip conditions, two different tips were used for the ‘writing’ measurement and the ‘retention’ measurement, which may result in some subtle differences between the data. But generally, in spite of the difference in scan step sizes and tip conditions, the concerned features of the region still demonstrate consistency before and after eight months.

2) As for the current images, by overlapping the two c-AFM images more precisely, it is shown

that the current signals (labelled with numbers) are still mostly consistent after eight months (Fig. A2). The main reason that current signals seem to be inconsistent previously is that the two c-AFM current images for ‘writing’ and ‘retention’ (Old Supplementary Fig. 13a, b) were not showing the exact same regions and the position offset is about tens of nanometers, indicated by the height images in previous Supplementary Fig. 13. Just as the Referee mentioned, since the size of the target region is only hundreds of nanometers, though the position is almost the same, it is not easy to find the exact same spot without any offset every time we measure. In addition, the arrows are drawn only for illustrating polarization directions and are not intentionally placed in the same position, which may also be misleading.

We have revised our previous Supplementary Fig. 13 (now Supplementary Fig. 14) by fixing the misalignment issue. All the images in panel b are shifted for the same amount of displacement in order to make sure the presented sample area is the same as that in panel a.

Figure A2 | Overlapping current images in Supplementary Fig. 14a (Writing) and Supplementary Fig. 14b (Retention). The overlapping shows consistency between the current signals measured before and after 8 months.

Figure S14 | Retention and erasure of the conductive DWs on 20 u.c. BTO/Si (Left panel for old version and right panel for new version). **a**, The written conductive DWs on 20 u.c. BTO/Si. **b**, Scanning of the created conductive DWs 8 months after creating it. Reading voltage is at +3 V. **c**, Erasing the created conductive DW by scanning over the whole region with the same negative voltage. From top to bottom: height image, LPFM phase image (arrows for polarization directions), c-AFM current image and phase-current overlapping image.

Q7: The authors mentioned an important point of these head-to-head domain walls in this study. That is the angular dependence of domain wall conducting behavior, which echoes my concerns. This delicate domain structure is not suitable for practical application. Can the authors show the angular dependency of domain wall conductance, such as the angle between polarization versus conducting current at a certain tip bias?

A1.3: Following Referee's suggestion, we conducted c-AFM measurement on a circular DW (Supplementary Fig. 11). The overlaying of c-AFM current image and LPFM amplitude image reveals that the H-H DW that is nearly perpendicular to the polarization direction shows conductivity. Such condition for conductivity may be demanding, but would in principle still be achievable by the four-terminal model discussed in **AI.1**. In addition, this can also be an excellent feature to be utilized to yield very high ON/OFF ratio and excellent turnability. Moreover, as BTO/Si is just an example of perovskite/Si prototype, other perovskites like the R-phase BiFeO₃ with ferroelectric polarizations lying along [111] and equivalent directions [Ref 28] has demonstrate less delicate conductivity, which can also be used for novel devices based on ferroelectric perovskite

oxide/Silicon heterostructures.

We have revised our manuscript based on the above discussion.

In the main text, Page 11 (text in blue): ‘Additionally, since the band bending and charge accumulation strongly depend on alignment of the DW with the polarization direction, only those H-H DWs nearly perpendicular to the polarization direction present prominent conductivity as depicted in Fig. 4e and further evidenced in Supplementary Fig. 11.’

Figure S11 | Conductivity of intrinsic circular DW on 20 u.c. BTO/Si. a,b, LPFM amplitude (a) and phase (b) images. Arrows demonstrate polarization orientations. **c,** c-AFM current image. **d,** Overlay of amplitude image and current image showing that H-H DW where DW is nearly perpendicular to polarization exhibits conductivity.

[28] Sharma, P, *et al.* Nonvolatile ferroelectric domain wall memory. *Sci. Adv.* **3**, e1700512 (2017).

REVIEWERS' COMMENTS

Reviewer #2 (Remarks to the Author):

The authors addressed the questions and concerns that I have.

Only one comment left:

The authors mentioned, "As to the size of the four-terminal model, this 4-terminal design is much smaller than the commercialized FRAM based on 1T1C structure ($\sim 0.5 \mu\text{m}^2$)". I have to remind the authors that to do a fair cell size benchmarking, one has to apply the same technology node. Physically, a four-terminal device cannot be smaller than a two-terminal device under the same process framework.

Responses to Reviewer #2:

The authors addressed the questions and concerns that I have.

We thank Referee #2 for the positive comments on our previous responses. Below we address the further one concern.

***Q1.1:** The authors mentioned, "As to the size of the four-terminal model, this 4-terminal design is much smaller than the commercialized FRAM based on 1T1C structure ($\sim 0.5 \text{ um}^2$)". I have to remind the authors that to do a fair cell size benchmarking, one has to apply the same technology node. Physically, a four-terminal device cannot be smaller than a two-terminal device under the same process framework.*

***A1.1:** We fully agree with the Referee about the physical size difference between the four-terminal and two-terminal devices. Nonetheless, this four-terminal design is still quite promising since it is already comparable or smaller than the commercialized 1T1C-type FeRAM. Also, as we have pointed out in our last response, the two-terminal design should work for BiFeO₃ and many other perovskite oxide membranes. For sure, these new designs will be explored in the future.*